# MAVEN: A Mesh-Aware Volumetric Encoding Network for Simulating 3D Flexible Deformation

**Zhe Feng**[1,*] **Shilong Tao**[2], **Haonan Sun**[2], **Shaohan Chen**[2], **Zhanxing Zhu**[3,†] **Yunhuai Liu**[2,†]

[1]Academy for Advanced Interdisciplinary Studies, Peking University
[2]School of Computer Science, Peking University
[3]School of Electrical and Computer Science, University of Southampton

## Abstract

Deep learning-based approaches, particularly graph neural networks (GNNs), have gained prominence in simulating flexible deformations and contacts of solids, due to their ability to handle unstructured physical fields and nonlinear regression on graph structures. However, existing GNNs commonly represent meshes with graphs built solely from vertices and edges. These approaches tend to overlook higher-dimensional spatial features, e.g., 2D facets and 3D cells, from the original geometry. As a result, it is challenging to accurately capture boundary representations and volumetric characteristics, though this information is critically important for modeling contact interactions and internal physical quantity propagation, particularly under sparse mesh discretization. In this paper, we introduce MAVEN, a **m**esh-**a**ware **v**olumetric **e**ncoding **n**etwork for simulating 3D flexible deformation, which explicitly models geometric mesh elements of higher dimension to achieve a more accurate and natural physical simulation. MAVEN establishes learnable mappings among 3D cells, 2D facets, and vertices, enabling flexible mutual transformations. Explicit geometric features are incorporated into the model to alleviate the burden of implicitly learning geometric patterns. Experimental results show that MAVEN consistently achieves state-of-the-art performance across established datasets and a novel metal stretch-bending task featuring large deformations and prolonged contacts. The code is available at https://github.com/zhe-feng27/MAVEN.

## 1 Introduction

Flexible deformation and contact of solids are prevalent across a wide range of real-world scenarios, ranging from industrial manufacturing (Tao et al., 2026; 2025a;b), aeronautical engineering (Phanden et al., 2021), to nuclear materials (Allen et al., 2012). Accurate modeling of these behaviors and their evolution significantly enhances the understanding of these scenarios. Although many classical numerical solvers, such as Finite Element Methods (Dhatt et al., 2012) and Material Point Methods (Bardenhagen et al., 2004), have been developed for solid systems, achieving the desired accuracy incurs high computational costs, as they rely on fine meshes and small time steps due to low-order approximations and the iterative solution of large linear systems. Recently, deep learning (DL) has rapidly emerged as a powerful tool for efficient physical simulation, showing great potential, particularly in molecular dynamics (Jumper et al., 2021), fluid simulations (Li et al., 2021), and structural solid deformations (Tang et al., 2026).

Among these DL-based solvers, graph neural networks (GNNs) have demonstrated superior performance in the domain of solid deformation, due to their ability to handle dynamic unstructured meshes and perform nonlinear regression on graphs (Sanchez-Gonzalez et al., 2020; Gao et al., 2022; Gladstone et al., 2024). To handle irregular solution domains, existing GNN-based methods input

---

*zhe.feng27@pku.edu.cn
†Corresponding authors.

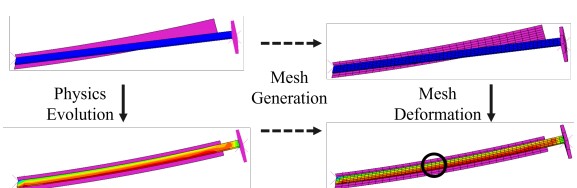
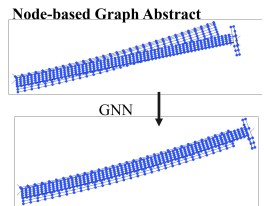

(a) Mesh division from original physical states. The purple region represents the fixed rigid body, while the blue region represents the deformable body subjected to clamping motion.

(b) Node-based graph construction

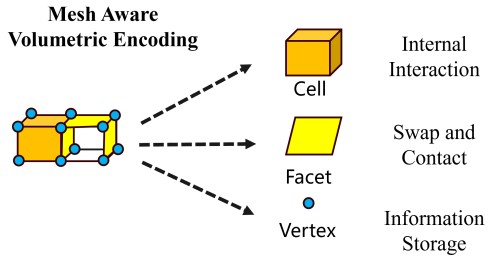
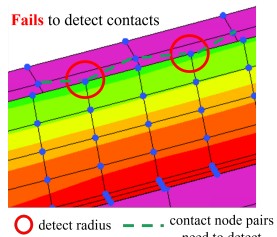

(c) Mesh aware volumetric encoding based on 3D cells, 2D facets

(d) Node-based method ignores contact features even with large radius

Figure 1: The physical state on the continuous material domain is discretized using structured meshes. Node-based methods construct point-edge graphs from the mesh and apply GNNs for computation. However, such abstraction may overlook contact interactions. A more effective approach should incorporate higher-dimensional geometric structures in the mesh, such as 3D cells and 2D facets, which retain accurate geometric information after discretization.

unstructured meshes, where the geometry is discretized into multiple connected simple cells using a regular polyhedron (Figure 1(a)). GNNs abstract mesh vertices into graph nodes to capture internal physical interactions, using edges defined by mesh connectivity, shown in Figure 1(b). Inter-object contact is typically detected via an interaction radius. Mesh edges serve as pathways for propagating physical quantities, making the mesh structure and its associated graph a central representation of the physical system.

Although GNNs are effective, their accuracy deteriorates on sparse meshes, which are commonly used for computation efficiency concerns in practice (Allen et al., 2023). As illustrated in Figure 1(d), the distance between the points in GNN differs from the actual distance between surfaces, and this deviation worsens under coarser mesh configurations. As a result, contact information may be missing without an appropriate detection radius. Increasing the radius may help, but with the cost of computations, there are still no guarantees of complete and accurate contact information. In addition, GNNs model internal propagation in approximating integral kernels (Anandkumar et al., 2020; Li et al., 2025), which is based on positional vectors and physical variables. However, with coarse meshes, nodes may not be sufficient to adequately sample neighborhood regions, hindering the accurate construction of characteristics of the localized physical field.

These limitations are mainly due to node-based modeling *by only using the vertices*. These methods represent meshes as graphs with edge features encoding distances, but existing approaches relying on topological representations often lose critical spatial features required for physical accuracy. Crucially, in addition to the vertices, the mesh contains a much more comprehensive set of high-dimensional geometric elements, that is, **2D facets** and **3D cells**, as illustrated in Figure 1(c), colored orange and yellow. *Our idea is that such high-dimensional elements like 2D facets and 3D cells could be incorporated to enable the model to better characterize geometric structures within 3D continuous space.* With this key incorporation, graph-based contact modeling can explicitly capture boundaries and contact as face-to-face geometries, making approaches more suitable for precise simulations. Additionally, even though in the case of coarse mesh discretization, the model might

lead to inaccurate integral approximations based on discretely sampled vertices, volumetric cells could compensate and maintain stable computations by retaining geometric continuity.

To fully exploit high-dimensional geometric elements in mesh-based neural networks, we design a novel framework within an encoder–processor–decoder architecture that explicitly embeds cell and facet elements into the model, thereby enhancing performance under sparse mesh conditions. Technically, we propose MAVEN, a model based on Mesh-Aware Volumetric Encoding, in which we construct explicit nodes for each geometric element in the mesh, including vertices, facets, and cells. During each processing step, the vertex information is encoded into higher-dimensional elements using learnable position-aware aggregators. The internal interactions and external loads are then handled at the cell level through facet nodes, allowing information to propagate through the mesh via its higher-dimensional structures (cells, facets). This cell-facet co-design enables comprehensive geometric modeling beyond node-based approaches. MAVEN achieves state-of-the-art performance in extensive evaluations. The main contributions of this paper can be summarized as follows.

- We propose a paradigm that explicitly incorporates high-dimensional geometric elements into the simulation of 3D solid systems. This approach enables the model to capture precise geometric information and maintain stability under sparse discretization conditions.
- We design MAVEN, a model based on mesh-aware volumetric encoding that captures high-dimensional geometries by explicitly modeling both cell and facet elements. MAVEN facilitates data transformation between elements through carefully designed transformation functions, and propagates information over a cell-facet graph using a modified two-stage message passing process.
- We compare MAVEN with state-of-the-art methods on public elastic deformation datasets and a metal bending problem, with elastic-plastic deformations and a coarse mesh. The experimental results demonstrate that MAVEN outperforms baselines in solid deformation with an acceptable computational efficiency.

## 2 RELATED WORK

### 2.1 LEARNING PHYSICAL SYSTEMS WITH GNNS

Recently, the application of Graph Neural Networks (GNNs) for simulating flexible dynamics has emerged as a promising research direction (Sanchez-Gonzalez et al., 2020; Han et al., 2022a; Shlomi et al., 2020; Gao et al., 2022). As a baseline and essential method, MGN (Pfaff et al., 2020) represents meshes as graphs by treating vertices as nodes and using connectivity and proximity-based edges, within and across objects. It adopts an Encoder-Processor-Decoder architecture, encoding relative positions as implicit geometric features and learning dynamics via message passing. Later studies primarily aim to improve message passing through more expressive architectures (Dwivedi & Bresson, 2021; Shao et al., 2022; Han et al., 2022b), use hierarchical graphs to propagate information in various ranges (Cao et al., 2023; Fortunato et al., 2022; Grigorev et al., 2023), and adopt a hybrid design that integrates both approaches (Yu et al., 2024). Since these methods model physical features only on vertices, we refer to them as **node-based GNN** approaches.

However, graph-based models often neglect essential high-dimensional geometry, particularly in sparse meshes common in real-world scenarios. For **inter-object contact**, true interactions occur between surfaces, yet node-based GNNs approximate them via vertex distances, causing errors when contact regions extend beyond vertex discretization (Allen et al., 2023). For **intra-object dynamics**, some studies (Li et al., 2025; Anandkumar et al., 2020) interpret message passing as an approximation of a local kernel function that performs discrete integration over information from neighboring graph nodes. In the sparse condition, meshes provide too few nodes to capture local geometry. Consequently, critical quantities such as volume and surface area are poorly estimated, and these errors propagate through message passing, leading to significant deviations in predictions.

### 2.2 GEOMETRIC ELEMENTS IN PHYSICAL SIMULATION

These limitations stem from node-based modeling that relies solely on mesh vertices, overlooking the rich set of high-dimensional geometric elements inherently present in the mesh. Mesh represen-

tations also include **3D cell** structures that accurately capture geometric partitioning in the continuous domain, and **2D facet** structures that define boundaries between regions and encode inter-object contact information. These elements contribute to more stable computations, particularly under sparse mesh conditions. For example, classical numerical methods (Reddy, 1993; Bardenhagen et al., 2004) describe physical quantities within volumetric elements by defining a family of shape functions (Zienkiewicz & Taylor, 2005) that interpolate physical qualities throughout the cell, and contact penalty terms are imposed on the integrals over the boundary facets. Based on this modeling approach, numerical methods can maintain controlled errors even under sparse mesh conditions.

A limited number of studies have focused on incorporating high-dimensional geometric information into DL–based physical simulations to improve computational accuracy. PHYMPGN (Zeng et al., 2025) follows discrete Laplace-Beltrami operators (Reuter et al., 2009), using face areas and cosine values of neighboring triangles for a single vertex as a broadcast operator between node pairs. This operator is limited to 2D settings and presents significant challenges when extending to 3D mesh domains. FIGNet (Allen et al., 2023; Lopez-Guevara et al., 2024) constructs face-to-face edges to capture contact relationships. Although effective for rigid bodies, these methods still face significant challenges in modeling internal propagation and dynamic deformations within 3D solids. To address this, we propose a novel DL-based architecture for 3D dynamic deformation simulation that intrinsically integrates high-dimensional geometric structures with hierarchical feature representations.

## 3 METHODOLOGY

### 3.1 PROBLEM DEFINITION

The evolution of a Lagrangian system is initiated by an initial material domain $\Omega_0$, together with a field function $\mathbf{U}(0, x)$ that defines the initial physical quantities, such as displacement, velocity, and pressure, at each material point $x$. At each time $t \in [0, T]$, we focus on the current physical state $\mathbf{U}(t, x)$ of every point $x \in \Omega_0$. To enable discrete computation, the initial material domain $\Omega_0$ is partitioned, without overlap or omission, into a set of regular tetrahedra (or hexahedra) cells $\mathcal{C}$, which collectively form the **mesh**. The collection of vertices and surface facets from these regular regions defines the set of vertices $\mathcal{V}$ and the set of facets $\mathcal{F}$ of the mesh, respectively. Physical field information $\mathbf{u}_i^t$ at time $t$ is stored at each vertex $v_i \in \mathcal{V}$ to approximate the continuous domain, allowing the state of any point of material to be estimated by interpolation from the values at the vertices of the corresponding cell. The exact shape of each cell and facet at any given time is determined by the current state of deformation of its vertices. Excessive distortion or even fracture may occur as a result. In this work, we primarily consider scenarios in which the mesh does not undergo severe distortion, which aligns with the assumptions commonly made in industrial simulation settings.

The simulation trajectory of a physical system originates from an initial physical field $\mathbf{u}^0$ governed by a discretization mesh $\mathcal{M} = \{\mathcal{C}, \mathcal{F}, \mathcal{V}\}$. The input of variable-time external forces $\{\mathbf{f}^0, \mathbf{f}^{\Delta t}, ..., \mathbf{f}^{K\Delta t}\}$ acting on the vertices drives the progression of the dynamic state, generating the sequence of physical evolution $\{\mathbf{u}^0, \mathbf{u}^{\Delta t}..., \mathbf{u}^{K\Delta t}\}$ in discretization evolution time $0, \Delta t, ..., K\Delta t = T$. The objective of the simulator is to predict next physical state $\mathbf{u}^{(k+1)\Delta t}$ from a history of previous states. In this paper, we consider $\{\mathbf{u}^0, \mathbf{u}^{(k-1)\Delta t}, \mathbf{u}^{k\Delta t}, \mathbf{f}^{k\Delta t}\}$ as input states. The rollout trajectory can be obtained by applying the simulator to the last prediction iteratively.

### 3.2 MODEL OVERVIEW

The overall architecture of MAVEN is illustrated in Figure 2, with the widely adopted encoder-processor-decoder framework (Pfaff et al., 2020). Unlike node-based models, MAVEN additionally models each element in the cell set $\{\mathcal{C}\}$ and facet set $\{\mathcal{F}\}$ as individual nodes participating in message passing, thus enhancing the ability to capture high-dimensional geometric information. First, MAVEN performs feature extraction for all node types. The cell and facet nodes are initialized with their geometric characteristics, such as 3D volume, surface area, as well as 2D area and perimeter, while the vertex nodes retain their physical quantities as input features. Subsequently, we employ a stack of processors to model physical interactions within and across solid objects. In each processor, the cell and facet nodes update their geometric representations from the nearby vertex nodes via a position-sensitive geometric aggregator. A modified two-stage message passing is then applied to propagate information during a cell-facet graph constructed by the relative geometric relationships

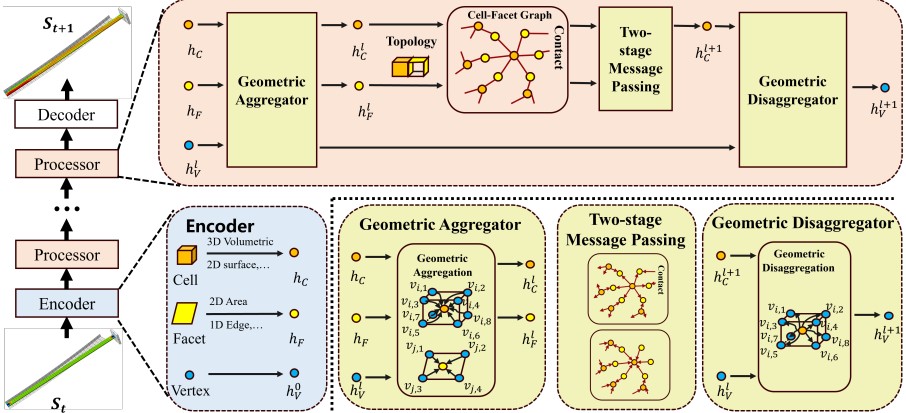

Figure 2: The overall structure of MAVEN. MAVEN follows an encoder–processor–decoder pipeline: it extracts geometric and physical features for vertices, cells, and facets, updates them through position-aware geometric aggregation and refined cell–facet message passing, and finally disaggregates the processed features back to vertices to produce smooth predictions.

between elements. Finally, a disaggregation operation distributes the aggregated features back to the vertex nodes, generating smooth intermediate results. The final processor output is subsequently mapped back to the original domain to produce the predicted results.

In MAVEN, cells and facets focus on different types of features. The facet is a pivotal hub for information exchange, where external forces, object contacts, and the propagation of internal physical quantities converge. The diverse information is integrated and subsequently transmitted back to their respective cells. Results(Allen et al., 2023) show that faces can capture contact information more effectively, which node-based models might otherwise neglect under sparse conditions. The cells fully characterize the geometric and volumetric properties of the 3D continuum domains. The adoption of volumetric features of adjacent cells as propagation coefficients for vertices significantly enhances the performance of 2D tasks (Zeng et al., 2025), motivating our design of the cell-facet propagation framework, ensuring comprehensive geometric information within the model.

Next, we elaborate on each key component of MAVEN. For convenience, in the following description we denote $\mathcal{A}$ as the feature fusion operator that integrates multiple features into a single representation, implemented via multilayer perceptrons (MLP) in practice. Various $\mathcal{A}$ are distinguished using the subscript and superscript notation.

### 3.3 ENCODER: GEOMETRY-INFORMED FEATURE ENCODING

The MAVEN encoder constructs feature representations for the cell, facet, and vertex nodes while also processing external force conditions and inter-facet contact relationships.

**Vertex node encoder** For a given vertex node $v_i \in \mathcal{V}$ and its associated physical field quantities $u_{v_i}^t$, we apply standard GNN practices to encode quantities into a high-dimensional latent space to derive the vertex node feature $\mathbf{h}_{v_i}^0$:

$$\mathbf{h}_{v_i}^0 = \mathcal{A}^{\mathcal{V}}(u_{v_i}^t) \tag{1}$$

**Cell and facet node encoder** Since cells and facets do not possess direct input features, we consider computing their representations from high-dimensional geometric properties. Inspired by the classical finite-volume method (Eymard et al., 2000), we posit that both the internal volume and surface area of a region are critical geometric descriptors. Accordingly, we use volume and total surface area as initial geometric features for each cell, while area and perimeter are used to characterize facet nodes. In addition, we incorporate the initial geometric attributes of each element to enhance the model's awareness of high-dimensional geometric variations over time. Let $\Omega(c_i), \Sigma(c_i)$ be the volume and surface area of cell $c_i$, and $\alpha(f_i), \lambda(f_i)$ be the area and perimeter of facet $f_i$, MAVEN

generates cell and facet features $\mathbf{h}_{c_i}$ and $\mathbf{h}_{f_i}$ as follows:

$$\mathbf{h}_{c_i} = \mathcal{A}^{\mathcal{C}}(\Omega(c_i^t), \Sigma(c_i^t), \Omega(c_i^0), \Sigma(c_i^0)), \quad \mathbf{h}_{f_i} = \mathcal{A}^{\mathcal{F}}(\alpha(f_i^t), \lambda(f_i^t), \alpha(f_i^0), \lambda(f_i^0)) \tag{2}$$

Here, all $\mathbf{h}_{v_i}^0$, $\mathbf{h}_{c_i}$, and $\mathbf{h}_{f_i}$ are projected to the same latent dimension, which is set to 128 in practice.

**Facet-to-facet features** Instead of constructing edges between vertices, MAVEN establishes contact connections directly between the interacting facets. We improve on (Allen et al., 2023) by applying a simplified Bounding Volume Hierarchy algorithm (Clark, 1976) to detect all pairs of faces within a collision radius $r$. For two contacting facets $f_s$ and $f_r$, the translation equivariant vectors of each face edge include: (1) the relative displacement between the center points $\mathbf{d}_{rs}^{\mathcal{F}} = \mathbf{p}_r - \mathbf{p}_s$ on the two faces, (2) the spanning vectors from the vertices of each face to the center of the other face $\mathbf{d}_{v_i}^{\mathcal{F}} = \mathbf{x}_{s_i} - \mathbf{p}_s$, $\mathbf{d}_{r_i}^{\mathcal{F}} = \mathbf{x}_{r_i} - \mathbf{p}_r$, and (3) the normal unit vectors of the face of the sender and receiver faces $n_s, n_r$. MAVEN generates face-to-face features $\mathbf{h}_{f_s \to f_r}$ as:

$$\mathbf{h}_{f_s \to f_r} = \mathcal{A}^{\mathcal{F} \leftrightarrow \mathcal{F}}([\mathbf{d}_{rs}^{\mathcal{F}}, [\mathbf{d}_{s_j}^{\mathcal{F}}]_{s_j \in f_s}, [\mathbf{d}_{r_j}^{\mathcal{F}}]_{r_j \in f_r}, \mathbf{n}_s, \mathbf{n}_r]) \tag{3}$$

**External force features** External forces are defined as scripted motions for specific surface vertices, which means that their next-step positions $\mathbf{x}^{t+1}$ are explicitly provided in the current step. We define the external force feature for each node $\mathbf{h}_{v_i}^S$ as its relative displacement to the next time step, and introduce a flag to indicate whether the node is scripted.

$$\mathbf{h}_{v_i}^S = \begin{cases} [1, \mathbf{x}_{v_i}^{t+1} - \mathbf{x}_{v_i}^t], & \text{if } v_i \text{ is scripted} \\ \mathbf{0}, & \text{if } v_i \text{ is not scripted} \end{cases} \tag{4}$$

In practice, scripted motions are typically applied over entire surface regions rather than isolated vertices, making it essential to impose constraints at the surface level. Therefore, we define scripted features on each facet $\mathbf{h}_{f_i}^S$ by concatenating motion-related features of all its associated vertex nodes.

$$\mathbf{h}_{f_i}^S = \mathcal{A}^S(\text{concat}_{v_j \in f_i}(\mathbf{h}_{v_i}^S)) \tag{5}$$

To ensure translational and permutation invariance, we sort the vertices of each facet by their distances to the facet centroid, thereby enforcing a unique representation.

## 3.4 PROCESSOR

All features extracted by the encoders are fed into a processor module composed of $L$ stacked layers. In the $l$-th layer, the processor takes the vertex features of the previous layer $\mathbf{h}_V^{l-1}$ and applies a geometric aggregator to incorporate the vertex information into the facet and cell nodes to generate geometric features $\mathbf{h}_C^l$ and $\mathbf{h}_F^l$. Two-stage message passing is used to propagate physical information across high-dimensional elements, where messages are first exchanged on facets and subsequently to cells. Finally, an inverse disaggregation decoder maps the updated cell-level features back to the vertex nodes for residual updates, producing spatially smooth features $\mathbf{h}_V^{l+1}$ over domain.

**Geometric Aggregator** Since vertex features are updated through the processor, it implies that the features of the cells and facets must also be updated. We update the features of each element by aggregating the features of all vertices that it contains. A straightforward approach is to concatenate the initial features with those of all associated vertices, followed by an aggregation operation. However, since each element contains a relatively large number of vertices (e.g., eight vertices in a hexahedron), this approach results in significant computational overhead. Another approach is to average the features of all vertices, following the conventional GNN. However, this leads to severe homogenization of features between vertices and overlooks the relative geometric relationships between the nodes in their corresponding cells.

Inspired by the shape function (Reddy, 1993) in numerical solvers, which describes physical fields in a cell using local coordinates, MAVEN constructs aggregation coefficients from each vertex to the element based on the local coordinate system of the element. These coefficients are shared across all processor layers. Let $\vec{\mathbf{d}}_{c_i, v_j}$ denote the position vector from the center of the cell $c_i$ to vertices $v_j \in c_i$, similar to $\vec{\mathbf{d}}_{f_k, v_l}$. Based on the local coordinate system, we employ an MLP to generate a centered normalized vertex aggregation weight $a_{c_i, v_j} \in \mathbb{R}$ for each cell.

$$a_{c_i,v_0}, \ldots, a_{c_i,v_{K-1}} = \text{MLP}\left(\underset{v\in\{v_0,\ldots,v_{K-1}\}}{\text{concat}}(\vec{\mathbf{d}}_{c_i,v})\right) \qquad \{v_0, \ldots, v_{K-1}\} \text{ represents } c_i \quad (6)$$

Similarly, coefficients $a_{f_i,v_0}, \ldots, a_{f_i,v_{M-1}}$ for each facet $f_i$ can be derived, where $K$ and $M$ denote the number of associated vertices for each cell and facet, respectively. The coefficients obtained are then utilized to perform weighted aggregation for element feature update. We also sort the vertices within each cell to ensure permutation invariance.

$$\mathbf{h}_{c_i}^l = \mathcal{A}_l^{\mathcal{V}\to\mathcal{C}}(\mathbf{h}_{c_i}, \sum_{v\in\{v_0,\ldots,v_{K-1}\}} a_{c_i,v}\mathbf{h}_v^l), \quad \mathbf{h}_{f_i}^l = \mathcal{A}_l^{\mathcal{V}\to\mathcal{F}}(\mathbf{h}_{f_i}, \sum_{v\in\{v_0,\ldots,v_{M-1}\}} a_{f_i,v}\mathbf{h}_v^l) \quad (7)$$

**Message passing on cell-facet graph** After extracting features, we construct a bipartite cell-facet graph $G = (V_G = \{\mathcal{C},\mathcal{F}\}, E_G)$ to explicitly capture the topological and geometric relationships between volumetric elements and their boundary interfaces. The $E_G$ contains all pairs $(c_i, f_j)$ for $f_j \in c_i$. MAVEN performs two distinct message passing steps, each dedicated to the facet and cell nodes, respectively. In the first stage, information is aggregated through facet nodes, which serve as 'edges', bridging not only adjacent cells but also mediating interactions between external forces or contacts and the internal dynamics of the object. Inter-object contact interactions are represented on the facet level, where information from all face-to-face edges is aggregated. To incorporate context from adjacent cells, we similarly employ a learnable aggregation scheme with $a_{f_i,c_j}$.

$$\mathbf{h}_{f_i}^{\mathcal{F}\to\mathcal{F},l} = \sum_{f_r=f_i} \mathcal{A}^{\mathcal{F}\to\mathcal{F}}(\mathbf{h}_{f_s\to f_r}, \mathbf{h}_{f_s}^l), \quad \mathbf{h}_{f_i}^{\to\mathcal{F},l} = \mathcal{A}_l^{\to\mathcal{F}}(\mathbf{h}_{f_i}^S, \mathbf{h}_{f_i}^{\mathcal{F}\to\mathcal{F},l}, \mathbf{h}_{f_i}^l, \sum_{(c_j,f_i)\in E_G} a_{f_i,c_j}\mathbf{h}_{c_j}^l)$$
$$(8)$$

In the second stage of message passing, each cell aggregates information from its associated facets. A similar strategy is adopted, employing symmetric aggregation coefficients $a_{c_i,f_j} = a_{f_i,c_j}$ to combine messages from multiple surfaces.

$$\mathbf{h}_{c_i}^{\to\mathcal{C},l} = \mathcal{A}_l^{\to\mathcal{C}}(h_{c_i}^l, \sum_{(c_i,f_j)\in E_G} a_{c_i,f_j} h_{f_j}^{\to\mathcal{F},l}) \quad (9)$$

**Geometric Disaggregator** Finally, the high-dimensional geometric information encoded in each cell is retransmitted to its associated vertex nodes using the same symmetric aggregation coefficients $a_{v_i,c_j} = a_{c_j,v_i}$. A residual connection is applied to update the vertex features. This disaggregation at the vertex level facilitates a boundary-aware averaging of cell-level information, contributing to globally smooth predictions across the domain.

$$\mathbf{h}_{v_i}^{\to\mathcal{V},l} = \mathcal{A}_l^{\to\mathcal{V}}(h_{v_i}^l, \sum_{v_i\in c_j} a_{v_i,c_j} h_{c_j}^{\to\mathcal{C},l}), \quad \mathbf{h}_{v_i}^{l+1} = \mathbf{h}_{v_i}^l + \mathbf{h}_{v_i}^{\to\mathcal{V},l} + \text{FFN}(\mathbf{h}_{v_i}^l + \mathbf{h}_{v_i}^{\to\mathcal{V},l}) \quad (10)$$

where $\mathbf{FFN}(\cdot)$ represents the feed-forward network used in the Transformer (Vaswani et al., 2017). The next layer uses $\mathbf{h}_{\mathcal{V}}^{l+1}$ as the input vertex features.

## 3.5 Decoder and Updater

Our model decodes the features of vertices $\mathbf{h}_{\mathcal{V}}^L$ using an MLP decoder to predict the velocity $\hat{\dot{x}}^{t+1}$ and the physical quantities $\hat{c}^{t+1}$ for the next state. The next position $\hat{x}^{t+1}$ is estimated by first-order integration $\hat{x}^{t+1} = \hat{\dot{x}}^{t+1} + x^t$

**Training Loss** We use the one-step MSE loss as a training objective. Since other physical quantities may be included, MSE in flexible dynamics is calculated as follows:

$$\mathcal{L} = \frac{1}{|\mathcal{V}|}\|x^{t+1} - \hat{x}^{t+1}\|^2 + \frac{1}{|\mathcal{V}|}\|c^{t+1} - \hat{c}^{t+1}\|^2 \quad (11)$$

## 3.6 Discussion

Here, we briefly discuss how MAVEN differs from existing approaches.

**Compared to classical FEM methods**, MAVEN learns complex, nonlinear physical behaviors directly from data, avoiding the hand-crafted constitutive models required in FEM. It generalizes across varying geometries and boundary conditions, enabling faster inference and improved scalability for large-scale simulations.

**Compared to basic DL methods such as MGN**, MAVEN propagates physical information through high-dimensional geometric elements, including cells and facets. This approach introduces a small amount of additional overhead. However, it enables MAVEN to accurately capture geometric information, which enhances the model's awareness of local neighborhoods and significantly improves its stability under sparse meshing conditions.

**Compared to hierarchical approaches**, MAVEN focuses mainly on accurately capturing local geometric details. Although hierarchical methods are effective at modeling long-range interactions, they offer limited benefits for precise geometric representation. Moreover, MAVEN can be readily extended to graphs after pooling through an automatic mesh, which we leave for future work.

**Compared to PhyMPGN**, which depends on the cotangent Laplacian for local geometry, MAVEN avoids the inherent limitations of Laplacian-based operators in 3D, where no symmetric, local, and linearly accurate purely geometric Laplacian exists (Wardetzky et al., 2007). By explicitly modeling high-dimensional geometric elements and constructing operators through message passing, MAVEN provides a more flexible geometric framework, better suited for 3D Lagrangian formulations, and can naturally adapt to other physical settings such as 3D Eulerian formulations.

**Compared to FIGNet**, which is tailored for rigid-body contact and lacks mechanisms for modeling volumetric physical propagation, MAVEN explicitly represents cells and performs message passing over higher-dimensional geometric elements. This allows MAVEN to capture intra-object dynamics with much higher fidelity, especially under sparse mesh. In our experiments, we treat FIGNet as an ablated variant using only facet-level information, while MAVEN's joint use of facets and cells yields better accuracy, underscoring the importance of modeling internal geometric structure.

## 4 EXPERIMENTS

**Datasets** To evaluate the efficiency of MAVEN, we test our model on datasets with different complexity in 3D solid simulation. The deforming plate (DP) (Pfaff et al., 2020) and cavity grasping (CG) (Linkerhägner et al., 2023) are typical public datasets for the autoregressive elasticity task. The DP dataset contains relatively **dense tetrahedral meshes**, while the CG dataset has **coarser meshes**. To further explore plastic scenarios, we establish a Metal Bending dataset (MBD) inspired by (Clausen et al., 2000) in real-world manufacturing, representing a class of solid deformation involving elastoplastic deformation, large displacements, and **very coarse** hexahedral meshes. In addition to target geometry, we also predict associated physical quantities in experiments, including inner stress (Stress) and equivalent plastic strain (PEEQ). The rollout steps for all datasets are restricted to between 75 and 125 for a consistent comparison. We briefly present the motion process of the dataset in Figure 3. See Appendix A for more details.

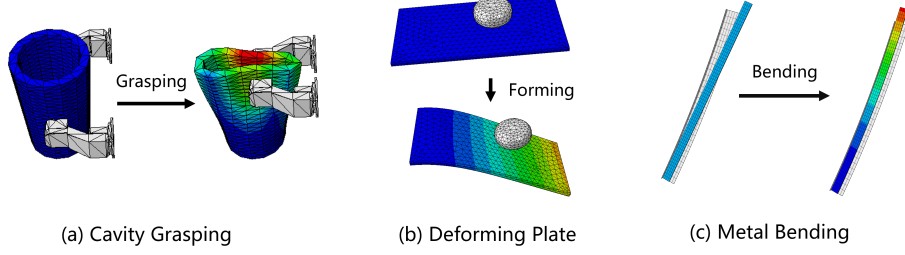

(a) Cavity Grasping         (b) Deforming Plate         (c) Metal Bending

Figure 3: Visual description of the dataset.

**Baselines** We comprehensively compare MAVEN against baselines with node-based graph simulators. These include classical node-based models MGN (Pfaff et al., 2020) and Graph Transformer (GT) (Yun et al., 2019), as well as hierarchical models HCMT (Yu et al., 2024) and HOOD (Grigorev et al., 2023). To further distinguish the functions between the cell and the facet elements, we

Table 1: Rollout results($\times 10^3$) for MAVEN and other baselines, with 50-step rollouts and full-sequence rollouts. Our results are derived by averaging the root mean square error (RMSE) values computed across all intermediate steps and test datasets.

| Model | Rollout | CG | DP | | MBD | | |
|---|---|---|---|---|---|---|---|
| | | Pos | Pos | Stress | Pos | Stress | PEEQ |
| MGN | 50 | $6.38_{\pm0.39}$ | $14.01_{\pm0.087}$ | $24,495,364_{\pm793,284}$ | $686.19_{\pm70.51}$ | $12508.07_{\pm392.56}$ | $0.24_{\pm0.045}$ |
| | ALL | $16.89_{\pm0.49}$ | $23.65_{\pm0.19}$ | $30,623,890_{\pm457,279}$ | $2012.16_{\pm299.60}$ | $9737.58_{\pm287.61}$ | $1.45_{\pm0.060}$ |
| GT | 50 | $6.15_{\pm0.37}$ | $14.72_{\pm0.30}$ | $24,384,076_{\pm915,030}$ | $678.87_{\pm39.99}$ | $10368.29_{\pm1737.58}$ | $0.41_{\pm0.060}$ |
| | ALL | $16.69_{\pm0.62}$ | $26.77_{\pm0.52}$ | $32,171,330_{\pm224,721}$ | $1406.61_{\pm72.84}$ | $14255.72_{\pm1203.51}$ | $2.07_{\pm0.083}$ |
| HCMT | 50 | $6.14_{\pm0.29}$ | $14.46_{\pm0.47}$ | $22,335,358_{\pm663,289}$ | $851.86_{\pm52.23}$ | $17940.56_{\pm2662.00}$ | $0.45_{\pm0.010}$ |
| | ALL | $16.87_{\pm0.24}$ | $24.94_{\pm0.76}$ | $30,317,188_{\pm457,279}$ | $2003.30_{\pm77.71}$ | $11539.27_{\pm834.80}$ | $1.30_{\pm0.16}$ |
| HOOD | 50 | $6.96_{\pm0.083}$ | $14.27_{\pm0.32}$ | $23,474,653_{\pm259,738}$ | $623.57_{\pm23.47}$ | $11739.02_{\pm982.37}$ | $0.37_{\pm0.078}$ |
| | ALL | $18.84_{\pm0.85}$ | $24.01_{\pm0.30}$ | $30,941,529_{\pm683,294}$ | $1762.41_{\pm35.92}$ | $8352.52_{\pm482.60}$ | $1.56_{\pm0.093}$ |
| FIGNet | 50 | $6.26_{\pm0.086}$ | $14.74_{\pm0.18}$ | $23,926,010_{\pm93,458}$ | $515.17_{\pm67.13}$ | $5583.71_{\pm1060.30}$ | $0.22_{\pm0.12}$ |
| | ALL | $17.59_{\pm0.51}$ | $26.51_{\pm0.23}$ | $31,491,198_{\pm237,542}$ | $1030.57_{\pm15.90}$ | $5402.31_{\pm805.40}$ | $1.09_{\pm0.52}$ |
| MAVEN | 50 | $\mathbf{5.21_{\pm0.0056}}$ | $\mathbf{13.78_{\pm0.17}}$ | $\mathbf{21,657,348_{\pm170,228}}$ | $\mathbf{276.73_{\pm44.46}}$ | $\mathbf{4901.56_{\pm46.06}}$ | $\mathbf{0.20_{\pm0.068}}$ |
| | ALL | $\mathbf{15.41_{\pm0.11}}$ | $\mathbf{23.41_{\pm0.32}}$ | $\mathbf{27,907,490_{\pm158,020}}$ | $\mathbf{810.42_{\pm24.08}}$ | $\mathbf{4776.72_{\pm71.20}}$ | $\mathbf{1.01_{\pm0.024}}$ |
| Improv. | | 13.07% | 1.33% | 5.49% | 33.82% | 11.90% | 8.67% |

adapt FIGNet (Allen et al., 2023) to propagate internal physical quantities through the vertices. The specific settings of the models are provided in Appendix C.

## 4.1 EXPERIMENTAL RESULTS

**Rollout Results** Table 1 shows the rollout results of all models, demonstrating that MAVEN consistently outperforms state-of-the-art methods to predict all physical quantities. From fine-grained to coarse meshes, MAVEN achieves incremental average improvements of $3.41\%$, $13.07\%$, and $18.13\%$ across the three datasets, respectively. This indicates that explicitly capturing high-dimensional geometric features is beneficial for physical simulation and becomes even more critical under sparser mesh conditions. In the MBD dataset, geometry-based methods (FIGNet, MAVEN) significantly outperform node-based approaches. In addition, the cell-element-aware architecture enables MAVEN to better capture variations in three-dimensional volumes compared to FIGNet.

**Visualization** Figure 4 presents the visualization results. Compared with other methods, MAVEN adopts a cell-based propagation approach, allowing physical contact information to be transmitted more effectively throughout the entire deformable body. As a result, MAVEN achieves lower errors even in regions that are far from the deformation part. At the same time, the use of facet-based contact detection allows MAVEN and FIGNet to maintain stable contact even under particularly coarse meshes. More visualization results and analyses can be found in Appendix E.

## 4.2 ABLATION STUDY

We validate two key components in our model, specifically, the aggregators that explicitly compute geometric features, and the feature aggregation method based on local geometric coordinates. We conducted systematic ablation studies on the MBD and CG to evaluate the contributions of different component models, testing three models: 1) Our full model; 2) **Model A**, which replaces geometric-based aggregation coefficients with a degree-averaging approach, is used to validate the effectiveness of our geometric aggregation strategy based on local coordinate systems; 3) **Model B** that replaces geometric input features of 3D cells and facets with zero padding, is used to assess the importance of explicitly computing geometric features; and 4) **Model C**, which removes explicit modeling of cell and facet nodes by averaging their precomputed geometric features onto neighboring vertex nodes and propagating them through standard message passing, is used to evaluate the role of explicitly representing higher-dimensional geometric elements.

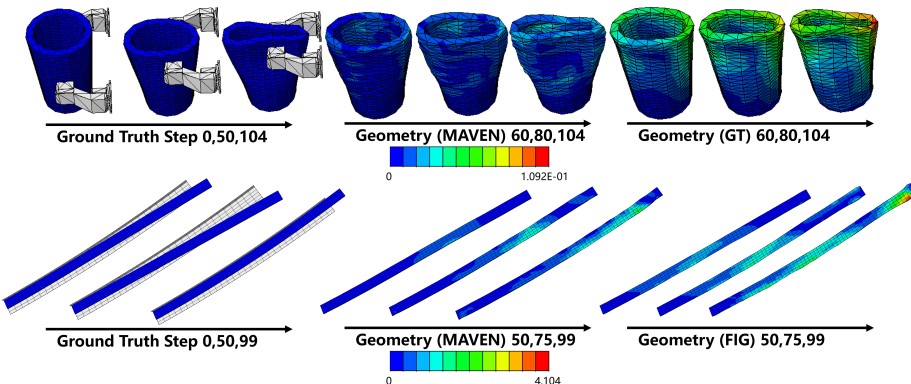

Figure 4: Visualization of error maps. The first and second rows respectively show sample visualizations from cavity grasping and metal bending datasets.

| Model | Rollout | CG | MBD | |
| --- | --- | --- | --- | --- |
| | | Pos | Pos | Stress |
| Ours | 50 | 5.21 | 276.73 | 4901.56 |
| | ALL | 15.41 | 810.42 | 4776.72 |
| Model A | 50 | 6.26 | 402.79 | 5302.67 |
| | ALL | 17.45 | 926.71 | 6683.94 |
| Model B | 50 | 5.41 | 391.40 | 6839.26 |
| | ALL | 15.933 | 1652.31 | 6680.39 |
| Model C | 50 | 6.05 | 632.71 | 9802.55 |
| | ALL | 17.08 | 1680.20 | 10375.86 |

Table 2: Ablation results on CG and MBD.

Table 2 shows the averaged error results in the test datasets. We observe that replacing geometry-aware aggregation with simple averaging (Model A) performs significantly worse on the generally sparse CG dataset. This indicates that such sparsity requires capturing detailed intra-element geometry to surpass standard node-based methods. Similarly, omitting explicit geometric features (Model B) leads to severe degradation on the highly sparse MBD dataset. This suggests that in extremely sparse settings, traditional GNNs struggle to infer local geometric structure implicitly and therefore fail to accurately capture the underlying geometric topology. For Model C, which does not explicitly model high-dimensional geometric elements, its performance on both datasets is close to traditional node-based methods. This suggests that adding geometric features alone, without modeling higher-dimensional topology, is insufficient for nodes to fully capture surrounding geometric structure.

These ablation results collectively demonstrate the effectiveness and necessity of MAVEN's design choices, including the explicit modeling of higher-dimensional geometric elements, the explicit computation of geometric features, and the use of geometry-aware aggregation for message updates.

## 5   CONCLUSION AND LIMITATIONS

We propose MAVEN, an architecture that models mesh geometry with high-dimensional features and learnable aggregation to simulate 3D solid contact and deformation on coarse meshes, outperforming baselines in physical propagation and contact representation. Despite these strengths, MAVEN remains sensitive to mesh quality due to its geometric modeling. In addition, as a local operator, it does not yet natively support efficient long-range interactions. Extending the framework to thin-shell, surface-based, or Eulerian systems also requires additional geometry-aware adaptations. Detailed discussion is provided in Appendix G.

## ACKNOWLEDGMENTS

This work is supported partly by the National Natural Science Foundation of China (NSFC) 62576013, National Key Research Plan under grant No.2024YFC2607404, the Jiangsu Provincial Key Research and Development Program under Grant BE2022065-1, BE2022065-3, and the Ningxia Domain-Specific Large Model Health Industry R&D No. 2024JBGS001.

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

## A  DATASET

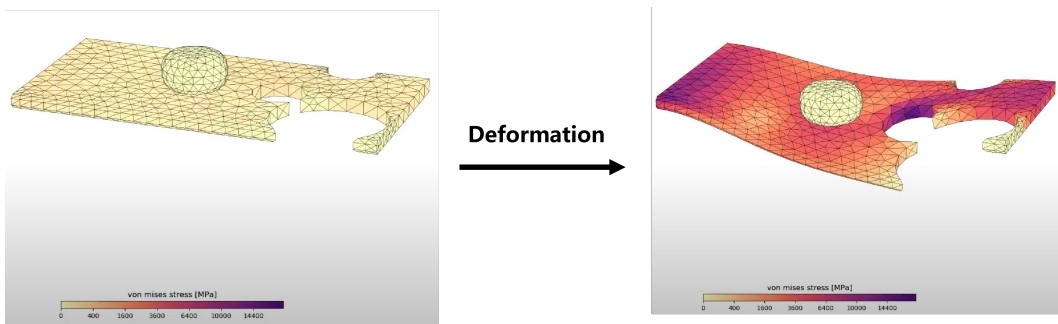

Figure 5: Description of Deforming Plate from (Pfaff et al., 2020).

**Deforming Plate Dataset** (Pfaff et al., 2020) This dataset contains 1,200 3D dynamic simulations of a hyperelastic deformable plate pressed by a rigid solid (Fig. 5). Each sample records the geometry of the plate and the internal stress, with an average of 1,271 points per simulation. In our setup, we unroll this dataset with a step size of 100 iterations to maintain consistent sequence lengths across all datasets. Following the protocol of the original article, we split the data into 1,000 training samples, 100 validation samples, and 100 test samples.

**Cavity Grasping** (Linkerhägner et al., 2023) The dataset comprises a three-dimensional dynamic simulation of deformable cavities subjected to gripping by a rigid gripper (Fig. 6), containing a total of 840 samples. The gripper is modeled as two rigid bodies, corresponding to its two jaws, which undergo motion in opposing directions. The deformable objects are cone-shaped cavities generated with randomly assigned radii in the range [50, 87.5]. Their material properties are specified as elastic, with Poisson's ratios drawn cyclically from the set -0.9, 0.0, 0.49. This dataset, primarily designed for autoregressive modeling tasks, provides temporal trajectories over 105 simulation steps. Each sample contains 1,386 points. Following the strategy of the original article, 600 samples are used for training, 120 for validation and 120 for testing.

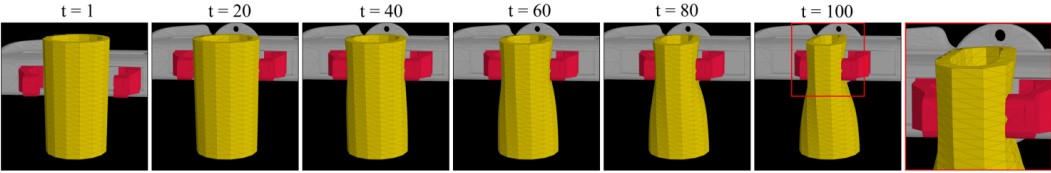

Figure 6: Description of Cavity Grasping from (Linkerhägner et al., 2023)

**Metal Bending Dataset** To rigorously validate the capability of MAVEN, we designed an industrially inspired test scenario featuring large deformations, coarse mesh discretization, and elastoplastic material behavior. This configuration mimics challenging real-world engineering applications such as metal-forming processes (Clausen et al., 2000), where computational methods must simultaneously handle geometric nonlinearity, material nonlinearity, and under-resolved meshes while maintaining physical accuracy and numerical stability. The results are calculated by ABAQUS software (Abaqus, 2011).

As illustrated in Fig. 7, this scenario involves clamping a straight aluminum profile using a specialized device, where the profile is first stretched beyond its elastic limit to induce plastic yielding, and then progressively pressed against a curved steel die through controlled displacement; the resulting compressive contact forces generate permanent plastic deformation to achieve the prescribed target geometry. The straight metal component is modeled as a slender component of size $2 \times 12 \times 200 mm^3$ to accurately replicate field conditions. For discretization, we employ a $1 \times 3 \times 5 mm^3$ mesh grid throughout the component. The material properties are configured as measured aluminum profile characteristics, with a Poisson's ratio of 0.37, Young's modulus of 69,000, and the true stress-strain curve depicted in Fig. 8(a). The 3D rigid die geometry is determined by its cross-sectional profile

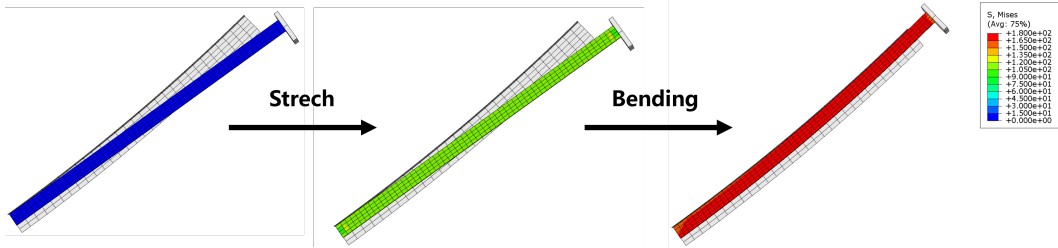

Figure 7: Description of Metal Bending.

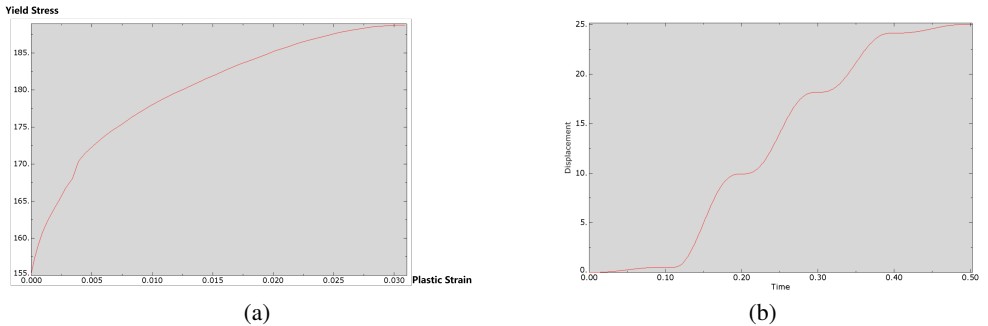

(a)                                           (b)

Figure 8: (a) True Stress-Strain Curve of Aluminum Profile. (b) Exemplar Motion Trajectory for Clamp Mechanism.

and a characteristic guiding curve. To ensure continuous contact between the aluminum workpiece and the die during bending operations, this guiding curve must maintain convexity and smoothness. We construct the curve by combining two circular arcs lying in the XY and XZ planes, respectively, each defining principal curvatures, and then synthesizing them into a composite spatial curve. In the ellipse $\frac{x^2}{a^2} + \frac{y^2}{b^2} = 1(a > b > 0)$ with a focal point at $(c, 0)$, we employ three distinct uniform distributions to control the morphology of the die, which are $\frac{c}{a} \sim \mathcal{U}[0.1, 0.3]$, $\frac{b}{a} \sim \mathcal{U}[0.1, 0.3]$ and $a \sim \mathcal{U}[170, 190]$(unit:mm). The clamping tool's motion trajectory is generated by applying a classical evolute algorithm along the characteristic curve (see Fig. 8(b) for an illustrative example). The dataset contains an average of 1163 nodes per sample, with rollout lengths varying between 75 and 125 timesteps. We generated a total of 1000 trajectories, divided into 800 for training, 100 for validation, and 100 for testing.

## B  METRICS

In our autoregressive framework, consistent with the baseline methods under comparison, we employ the Root Mean Square Error (RMSE) as the evaluation metric. Given the predicted physical quantity $\hat{y}_i$ and ground truth value $y_i$ of current $N$ vertices, the RMSE is calculated as:

$$RMSE = \sqrt{\frac{1}{n}\|\hat{y}_i - y_i\|^2} \tag{12}$$

For positions, we take the distance between two points as the sole comparative physical quantity. To evaluate the performance across multiple trajectories and frames, we report the final error metric as the average value computed over all frames for each individual trajectory:

$$ERROR = \frac{\sum_{i=1}^{M} \sum_{j=1}^{T_i} RMSE_{i,j}}{\sum_{i=1}^{M} T_i} \tag{13}$$

The number of trajectories is denoted by $M$, where $T_i$ represents the length (number of timesteps) of each individual trajectory.

## C  IMPLEMENTATION

Table 3: Key hyperparameters and parameter numbers of models.

| | | Deforming Plate | Metal Bending | Cavity Grasping |
|---|---|---|---|---|
| MGN | HIDDENS LAYERS | [128, 128] 15 | [128, 128] 15 | [128, 128] 15 |
| | PARAMETERS(M) | 3.85M | 3.85M | 3.85M |
| HCMT | HIDDENS LAYERS | [144, 144] 12+3 | [144, 144] 6+9 | [144, 144] 10 + 5 |
| | PARAMETERS(M) | 3.24M | 3.24M | 3.11M |
| FIGNet | HIDDENS LAYERS | [96, 96] 15 | [96, 96] 15 | [96, 96] 15 |
| | PARAMETERS(M) | 3.08M | 3.48M | 3.05M |
| HOOD | HIDDENS LAYERS | [128, 128] 12+3 | [128, 128] 6+9 | [128, 128] 10+5 |
| | PARAMETERS(M) | 3.56M | 3.16M | 3.47M |
| GT | HIDDENS LAYERS | [128, 128] 15 | [128, 128] 15 | [128, 128] 15 |
| | PARAMETERS(M) | 3.64M | 3.61M | 3.57M |
| MAVEN | HIDDENS LAYERS | [96, 96] 15 | [96, 96] 15 | [96, 96] 15 |
| | PARAMETERS(M) | 3.11M | 3.15M | 3.08M |

**Model Hyperparameters** To ensure a fair comparison, all models were evaluated under similar parameter budgets and computational costs, with detailed configurations provided in Table 3. Due to the larger parameter count per block in MAVEN and FIGNet, these models employed fewer propagation layers. For HCMT and HOOD, while the optimal hierarchy layer count originally reported was 5 layers on the deforming plate dataset, we reduced it to 3 layers in our implementation because the hierarchical partitioning algorithm failed to correctly subdivide new meshes beyond the fifth layer. For all datasets, we adopted a uniform batch size of 8 in all experiments.

**Training Implementation** All models were uniformly trained in 1M steps. We used Adam optimizer (Kingma, 2015), with a learning rate decreasing from $10^{-4}$ to $10^{-5}$. All experiments were conducted using a single RTX 3090 24GB GPU and repeated three times for the calculation of the standard deviation. All models were trained using Mean Squared Error (MSE) as the loss function, where each physical quantity was first normalized via Gaussian standardization and then summed directly to compute the final loss. This standardized approach ensured a fair and controlled comparison to objectively assess the relative performance of each method.

**Dataset Input and Detect Parameter** The inputs and outputs of each dataset, as well as the corresponding node-based contact detection radius $r_W$ and face-based contact detection radius $r_F$, are shown in Table 4. To ensure fairness, the node-based model and our model are able to detect nearly the same number of contact edges.

Table 4: Model input, output and contact detection parameters for dataset. $S$ denotes stress, and $P$ denotes PEEQ.

| Dataset | Input | Output | $r_W$ | $r_F$ | noise |
|---|---|---|---|---|---|
| Deforming Plate | $type_i, x_i^t - x_i^{t-1}, S_i^t$ | $x_i^{t+1} - x_i^t, S_i^{t+1} - S_i^t$ | 0.03 | 0.01 | 0.003 |
| Cavity Grasping | $type_i, x_i^t - x_i^{t-1},$ | $x_i^{t+1} - x_i^t$ | 0.1 | 0.05 | 0.01 |
| Metal Bending | $type_i, x_i^t - x_i^{t-1}, S_i^t, P_i^t$ | $x_i^{t+1} - x_i^t, S_i^{t+1} - S_i^t, P_i^{t+1} - P_i^t$ | 1 | 0.3 | 0.1 |

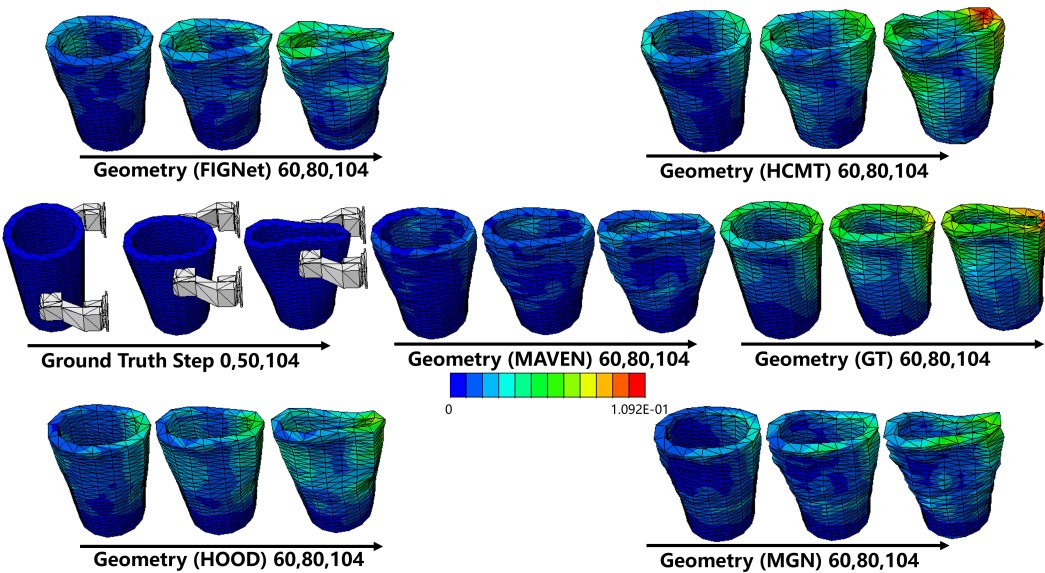

Figure 9: Visualization of distance error map on Cavity Grasping.

# D  MODEL EFFICIENCY

We briefly discuss the computational efficiency of MAVEN. Table 5 shows the runtime performance of various baselines on two datasets. The computational overhead of MAVEN primarily stems from geometric feature computation and inter-node mapping operations, particularly on hexahedral meshes. However, MAVEN still achieves an efficiency improvement of $2922.66\%$ over the Abaqus simulators ($712.44$ms per step) in the metal bending dataset.

Table 5: The inference time per step(ms) for each model on three dataset

| Dataset | MAVEN | MGN | FIGNet | HCMT | HOOD | GT |
|---------|-------|-----|--------|------|------|-----|
| CG | 25.48 | 18.37 | 24.15 | 55.05 | 51.37 | 22.02 |
| DP | 48.27 | 39.73 | 43.08 | 64.94 | 57.28 | 54.83 |
| MBD | 23.57 | 17.42 | 17.08 | 43.68 | 36.45 | 22.34 |

# E  VISUALIZATION

Figs. 9 and 10 present the complete visualization results for each dataset. It is worth noting that in the Metal Bending dataset, message-passing-based methods MGN and HOOD exhibit severe mesh distortions, especially at contact regions. In contrast, graph-attention-based methods HCMT and GT better preserve mesh shapes, though their overall structures become distorted. However, all of these methods suffer from severe interpenetration issues, as further illustrated in the rollout animations provided in the Supplementary Material. We believe that this phenomenon arises because individual nodes cannot accurately identify their intended contact regions. Enlarging the detection radius introduces many irrelevant points into the contact set, which, to some extent, is also exacerbated by the large discrepancies in mesh lengths across the x, y, and z axes. More specifically, near the fixed end, the mesh exhibits severe interpenetration. Closer to the moving end, the deformable body is heavily constrained by the lower rigid body, preventing it from generating the correct motion and causing it to stagnate. This indicates that facet-based contact detection is of critical importance when dealing with sparse meshes.

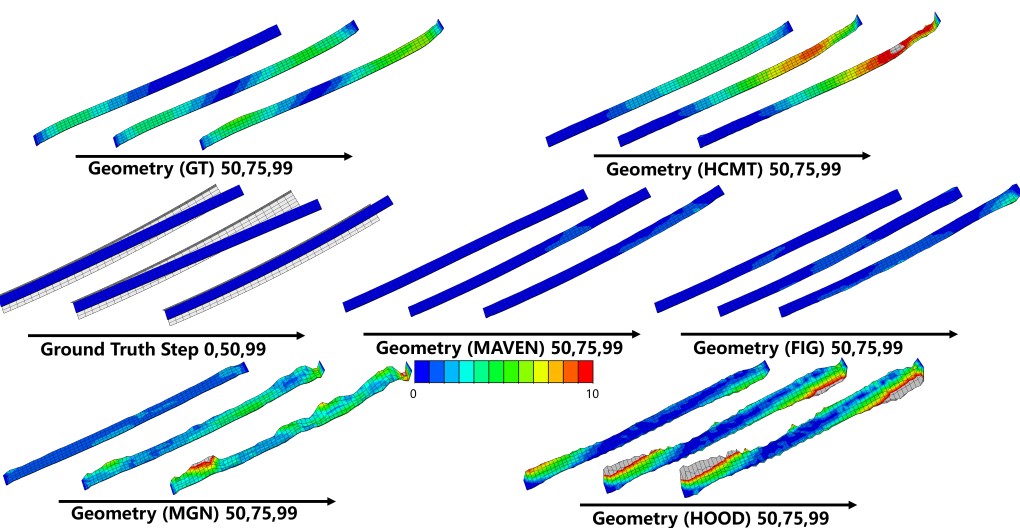

Figure 10: Visualization of distance error map on Metal Bending dataset. Gray indicates that the error at this location exceeds the given error upper bound.

## F   STATEMENT ON THE USE OF LARGE LANGUAGE MODELS

We declare that LLMs are mainly used in this paper to improve the clarity and fluency of the text, and to a limited extent for code generation in technically mature modules, to reduce repetitive work.

## G   LIMITATION AND FUTURE WORK

In this section, we briefly discuss the limitations of MAVEN and outline several promising directions for future work.

**The dependence on mesh quality**. Since MAVEN explicitly models cell-level information, the geometric quality of the initial mesh can substantially affect its performance. This behavior resembles that of traditional numerical methods (e.g., FEM, FVM) rather than node-based GNN models. In our experiments on the Cavity Grasping dataset, we compared meshes generated by a basic triangulation procedure with those produced by a higher-quality meshing algorithm. The results show that MAVEN's performance degrades markedly when operating on low-quality meshes. However, most existing deep learning datasets do not provide their original mesh discretizations, and meshes reconstructed from point clouds typically exhibit very low geometric quality. Consequently, generating more datasets with high-quality meshes, as well as reducing sensitivity to mesh quality, constitutes an important direction for our future work.

**Support long-range interaction.** While MAVEN is fundamentally designed as a local operator that explicitly leverages high-dimensional geometric structures (facets and cells), it does not yet natively capture long-range interactions. Existing graph pooling strategies (e.g., kNN clustering, BFS grouping, global slicing) can, in principle, be directly applied to MAVEN cell–facet graph to expand the receptive field. However, these methods often introduce significant computational overhead and offer limited benefit in modeling a global high-dimensional geometric structure. Developing a geometry-aware hierarchical extension that supports efficient long-range information propagation therefore represents an important direction for future work.

**Extend to boarder range of systems.** MAVEN is designed primarily for elastoplastic solid simulation on arbitrary 3D volumetric meshes, where explicit modeling of facets and cells provides high geometric fidelity. Although MAVEN can be adapted to thin-shell or surface-based geometries through appropriate redefinition of geometric features, and can extend to Eulerian formulations using fixed meshes and standard boundary-condition encodings, these adaptations require additional geometry-aware considerations and suitable datasets. Extending MAVEN into a unified framework

capable of supporting surface-based systems, thin-shell structures, and Eulerian physical simulations on sparse and irregular 3D meshes remains an important direction for future work.

