# OpenReview forum: "MAVEN: A Mesh-Aware Volumetric Encoding Network for Simulating 3D Flexible Deformation"
_ICLR.cc/2026/Conference — ICLR 2026 Poster_

### Official Review · Reviewer_3Wkt · 2025-10-27

**Soundness:** 2
**Presentation:** 2
**Contribution:** 2
**Rating:** 6
**Confidence:** 4

**Summary:**

MAVEN explicitly models high-dimension geometric mesh elements for physical simulation.

**I am not an expert in this field, so an accurate evaluation is difficult for me. I will defer to the assessment of other reviewers.**

**Strengths:**

- It is important to utilize more diverse 3D information when performing mesh-based simulation.

- The work demonstrates both accuracy and computational efficiency; despite using more information, the computational efficiency did not significantly degrade.

**Weaknesses:**

- As the authors themselves mentioned in the Related Works section, existing studies have also used high-dimensional information such as Cell.

**Questions:**

See Weakness section

---

> ### Author Response · Authors · 2025-11-23
> **Response 1**
>
> We sincerely appreciate your detailed feedback and the effort devoted to evaluating our paper. Below, we offer our responses to your comments.
>
> > Existing studies have also used high-dimensional information
>
> We appreciate you for pointing out that our comparison with other geometry-based models was not sufficiently detailed. We have added a more detailed comparison and discussion between MAVEN and prior approaches that incorporate higher-dimensional geometric information, including PhyMPGN [1] and FIGNet [2], in our revision manuscript(Section 3.6).
>
> - **PhyMPGN** employs the cotangent Laplacian operator to model physical dynamics, utilizing local geometric information around each node for message passing. However, for **3D problems**, there is **no Laplacian operator** that is simultaneously symmetric, local, and linearly accurate while being purely geometric and free of shape functions [3]. In contrast, MAVEN provides a **more flexible framework** for leveraging geometric information by explicitly modeling high-dimensional geometric elements and constructing operators through message passing. This design **enables MAVEN to better address the challenges inherent to 3D Lagrangian formulations**, and it can naturally adapt and potentially transfer to other physical settings, such as 3D Eulerian formulations. These properties demonstrate the generality and applicability of MAVEN across a wide range of physical simulation scenarios.
> - **FIGNet** is specifically designed for rigid-body contact problems, and its architecture does not include mechanisms for modeling **the propagation of physical fields within the interior of objects**. As a result, it primarily focuses on surface-level interactions rather than volumetric dynamics. In comparison, MAVEN explicitly represents the **internal geometric decomposition through cells** and performs message passing over these higher-dimensional elements. This enables MAVEN to capture intra-object physical propagation with significantly higher fidelity, particularly in settings where the geometric structure is sparse. In our experimental setup, we treat FIGNet as a special ablation variant that relies solely on facet-level information. The experimental results show that MAVEN, which jointly **incorporates both facet and cell representations**, achieves **substantially higher accuracy**. This highlights the importance of explicitly modeling internal geometric elements for capturing volumetric physical dynamics.
>
> In summary, MAVEN offers a **flexible and efficient framework** that integrates **both surface and volumetric geometric information**, enabling broader applicability across diverse 3D physical simulation scenarios.
>
> Sincerely thank you again for your insightful feedback. Should you have any additional questions or concerns, we would be pleased to provide further clarification and engage in continued discussion.
>
> [1] Bocheng Zeng, et al. PhyMPGN: Physics-encoded message passing graph network for spatiotemporal PDE systems. In *The Thirteenth International Conference on Learning Representations*, 2025.
>
> [2] Kelsey R. Allen, et al. Learning rigid dynamics with face interaction graph networks. *arXiv preprint* arXiv:2212.03574, 2022.
>
> [3] Wardetzky, Max, et al. “Discrete Laplace operators: no free lunch.” *Symposium on Geometry Processing*. Vol. 33. 2007.

---

### Official Review · Reviewer_ZTpm · 2025-10-29

**Soundness:** 2
**Presentation:** 2
**Contribution:** 3
**Rating:** 6
**Confidence:** 4

**Summary:**

The proposed paper, MAVEN, presents a graph neural network (GNN) model for simulating physics, particularly flexible solid deformations, including contacts. The key contribution of the work is the incorporation of additional explicit geometric features (cells and faces) into mesh graph network line of simulators, which previously considered only vertices and edges. This design reduces the burden of implicitly learning geometric patterns, enabling the network to capture 3D spatial structure and behaviour better.

The authors describe how to integrate these higher-order geometric entities using geometric aggregators and disaggregators, along with additional message passing on cell–facet graphs. MAVEN is compared against MeshGraphNets and several follow-up models in this category. It consistently achieves state-of-the-art performance across a few established datasets and on a new metal stretch–bending benchmark that features large deformations and sparse meshes.

**Strengths:**

1.	The core idea that incorporating higher-dimensional elements, such as 2D facets and 3D cells, enables better geometric representation of volumetric solids is a strong conceptual insight.

2.	The paper carefully explains how to integrate these geometric features, introducing geometric aggregators/disaggregators and modified message-passing schemes. The authors analyse design choices such as averaging versus learning coefficients in a local coordinate system and demonstrate where this new model is most beneficial (for coarse mesh discretisation and contact scenarios). Overall, it extends the expressive capacity of GNNs for physical simulation without much computational cost.

3.	The draft is clearly written. It systematically and thoroughly investigates different aspects of the idea. Considering that GNN-based simulation has received qwuite a bit of traction, this work provides nice insights, valuable to this community.

**Weaknesses:**

1.	It is unclear whether MAVEN can handle beyond volumetric solids. MeshGraphNets can model a wider range of systems, such as cloth (thin shell) or fluid dynamics, whereas MAVEN seems focused only on deformable solids. It would be useful to clarify whether MAVEN can handle surface-based or thin-shell geometries (such as cloth) and whether a version using only faces, without volumetric cells, is feasible. This would also make the comparison with surface-based GNNs such as HOOD fairer.

2.	The discussion focuses exclusively on Lagrangian (mesh-based) systems. The authors should comment on how MAVEN could extend to or differ from Eulerian formulations. Eulerian systems are explored in the original MeshGraphNet and its variants, but MAVEN's volumetric encoding will prevent GNN simulation in such scenarios.

3.	The paper would benefit from positioning the approach relative to Physics-Informed Neural Networks (PINNs), see [Karniadakis et al.]

[Karniadakis et al.] - Karniadakis, George Em, et al. "Physics-informed machine learning." Nature Reviews Physics 3.6 (2021): 422-440.

Minor comments and suggestions

1.	The term *flexible deformations* (used in the title and abstract) is somewhat vague. Consider using elastic or soft-body deformations, which more accurately describe 3D deformable solids.

2.	The overview figure’s caption could be expanded to explain the method flow better.

3.	There are a few typos:
a.	L292 likely refers to a geometric aggregator rather than an encoder.  b.	In Figure 4, the label GT in the top-right should probably read FIG.

**Questions:**

1.	How can MAVEN handle surface meshes, such as thin shells or 2D manifolds, without volume or Eulerian systems? Could the model be extended to such cases?

2.	Around L285/L318, the paper states: “To ensure translational and permutation invariance, we sort the vertices of each facet by their distances to the facet centroid, thereby enforcing a unique representation.” It is not clear how this sorting leads to invariance. Please provide a detailed justification.

3.	What are the optimisable parameters in the method? In Section 3.5, a list of additional optimisable parameters or MLPs compared to the baseline method (MGN) could be added.

---

> ### Author Response · Authors · 2025-11-23
> **Response 1**
>
> We greatly appreciate your thoughtful and constructive comments. Your feedback has been instrumental in refining both the presentation and technical depth of our paper. Below, we provide our detailed responses to the points you raised.
>
> > It would be useful to clarify whether MAVEN can handle surface-based or thin-shell geometries (such as cloth) and whether a version using only faces, without volumetric cells, is feasible. This would also make the comparison with surface-based GNNs such as HOOD fairer.
>
> We thank the reviewer for raising this question regarding MAVEN’s ability to handle surface-based or thin-shell geometries (e.g., cloth) and the feasibility of a faces-only (no volumetric cells) version. We will address this concern in three points.
>
> - **Thin-shell geometry as volumetric approximation.** In engineering applications such as cloth simulation, a thin-shell structure refers to a continuum model whose thickness is much smaller than its in-plane dimensions. The material is typically assumed to have uniform thickness, and the model is used to capture key mechanical behaviors including bending, stretching, and shear. Under this interpretation, the facets in MAVEN can be viewed as representing both sides of each surface patch, along with the effective contact surface implied by the neglected thickness of a thin shell. The cells, in turn, correspond to the complete geometric information of each surface patch. With this simple adaptation, MAVEN can be effectively applied to thin-shell geometries.
> - **MAVEN focuses on volumetric solids.** Full 3D elastoplastic simulation is generally regarded as substantially more complex both physically and numerically than thin-shell or cloth dynamics [1], due to its strong **material nonlinearities** (e.g., yielding and hardening), the presence of **large finite strains**, and the need to accurately handle **volumetric contact interactions**. These factors greatly increase modeling and computational difficulty, which motivates our emphasis on this more demanding domain that MAVEN is specifically designed to address.
> - **The need for geometry-aware modeling reflects MAVEN’s higher geometric fidelity.** Indeed, the fact that MAVEN requires slight adjustments of geometric features to accommodate different geometric regimes demonstrates that the method aims to **accurately capture underlying geometric structures**, rather than collapsing all meshes into a uniform point-edge graph for many existing GNNs [2, 3]. While this design grants MAVEN stronger geometric expressiveness, it also means that supporting fundamentally different systems requires explicit geometric considerations. To maintain clarity and consistency in method presentation, the current paper focuses on elastoplastic solid mechanics.

---

> ### Author Response · Authors · 2025-11-23
> **Response 2**
>
> > How MAVEN could extend to or differ from Eulerian formulations
>
> We appreciate the reviewer’s interest in extending MAVEN to broader physical settings, particularly those involving Euler formulations. Below, we provide a detailed discussion on this point.
>
> - **MAVEN can naturally support physical simulations under Euler formulations.** Similar to the Lagrangian setting, Euler formulations also require a mesh to describe the computational domain of interest. One difference is that, during the simulation, the mesh in the Eulerian view typically remains fixed in shape. As a result, the facet and cell elements in MAVEN can be defined in a straightforward manner. In this case, the geometric attributes of facet and cell nodes remain constant throughout the autoregressive process, while the physical field features associated with their neighboring nodes evolve over time. Another consideration in the Eulerian setting is the treatment of boundary conditions, which may include Dirichlet (used in MGN), Neumann, and Robin conditions. Dirichlet conditions can be embedded directly with our current approach. For Neumann and Robin conditions, existing boundary-condition encoding techniques [4, 5] can be integrated to extract the corresponding features and replace the scripted features used in our implementation. Overall, these adaptations allow MAVEN to extend naturally to Euler-based physical simulations.
> - **MAVEN is primarily designed for physical simulation on arbitrary 3D meshes. **While existing mesh-based methods have shown progress on uniform grids in 2D and 3D[6], or on arbitrary meshes in 2D[5, 7], **only a few general-purpose approaches are available for arbitrary 3D meshes** (as noted in our response to the first question). By explicitly modeling higher-dimensional geometric elements such as facets and cells, MAVEN is able to represent complex 3D geometries more accurately. Our experiments, which span a variety of elastoplastic deformation and contact problems on heterogeneous and irregular 3D meshes, demonstrate the advantages of MAVEN, particularly **in sparse and highly non-uniform geometric settings**. Unfortunately, to the best of our knowledge, there is no publicly available dataset that provides sparse and arbitrary 3D geometries under Euler formulations that would be suitable for our experimental setting. We would greatly appreciate it if the reviewer could point us toward any suitable datasets. Extending MAVEN to broader physical simulation settings remains an important direction for future work, and we would be glad to explore such datasets if available.
>
> Overall, these considerations explain why the present work concentrates on elastoplastic solids, while leaving other geometric regimes as promising directions for future exploration. We have also incorporated these extensions into the revised manuscript(Appendix G).

---

> ### Author Response · Authors · 2025-11-23
> **Response 3**
>
> > Positioning the approach relative to Physics-Informed Neural Networks (PINNs)
>
> While we appreciate the strong performance of PINNs in certain settings, particularly well-posed PDE problems with known governing equations, **they are not directly applicable to the scenarios studied in our work**. We did not include PINNs as baselines for the following reasons:
>
> - **Dependence on explicit PDE forms.** PINNs require knowledge of the exact governing PDEs in order to embed them into the loss function. Our dataset involves complex elasto-plastic behavior with **heterogeneous, nonlinear, and strongly coupled mechanical processes**, including material yielding and moving contacts. It is **not feasible to specify a single global PDE** (such as the Navier–Stokes equations in standard CFD) that accurately captures the behavior of the entire system.
> - **Instance-specific training.** Most PINN-based solvers, including Pirate PINN, are designed to solve **one PDE instance per training run**, which means they must be retrained whenever the geometry, material parameters, or boundary conditions change. In contrast, our goal is to learn a **generalizable solver** that can operate across large-scale datasets where **each sample corresponds to a different PDE instance**. This difference in problem formulation makes PINNs unsuitable as a fair baseline for our learning setting.
> - **Assumptions on domain representation.** Many PINN methods assume **regular spatial domains and Eulerian formulations**, which is also the case for Pirate PINN. Our benchmark tasks involve **irregular geometries**, **Lagrangian discretizations**, and **dynamically evolving contacts**, which violate these assumptions and would require substantial modifications for PINNs to be applied in a meaningful way.
>
> In summary, while PINNs are effective for solving single-instance PDEs with known and well-defined physics, they differ **significantly in scope, assumptions, and applicability** from the problem setting considered in our work. For these reasons, we do not include them as baselines. We agree that physics-informed and data-driven approaches are complementary, and exploring their integration remains an exciting direction for future research.

---

> ### Author Response · Authors · 2025-11-23
> **Response 4**
>
> > Other comments, suggestions and questions
>
> Thank you for highlighting the aspects of the manuscript that could be improved in terms of clarity. In the revised version(Figure 2, Section 3.3), we have refined and clarified the previously ambiguous parts to enhance the overall presentation.
>
> 1. **Term** **flexible deformations**.  We appreciate your suggestion regarding improving the rigor of our scenario description. Our use of the term “flexible” was intended to refer to a class of solid mechanics settings involving uncertain contacts and large deformations. If the paper is accepted, we will incorporate the term “soft-body” into the description to further clarify the scope of the work.
> 2. **Expand overview figure’s caption**. MAVEN follows an encoder–processor–decoder pipeline: it extracts geometric and physical features for vertices, cells, and facets, updates them through position-aware geometric aggregation and refined cell–facet message passing, and finally disaggregates the processed features back to vertices to produce smooth predictions. This clarification has already been incorporated into the revised version of the manuscript.
> 3. **Possible typos**. We appreciate the reviewer for pointing out the inappropriate use of the term “geometric encoder,” and we have corrected this in the revised version of the manuscript. For the result visualizations, we selected different baselines for the two datasets to illustrate **how different types of models exhibit distinct behaviors on each dataset**. In Appendix F, we provide visualizations of all baselines for both the Cavity Grasping and Metal Bending datasets, offering a complete comparison that highlights the learning differences among the various model categories.
> 4. **Ability of MAVEN on surface meshes without volume or Eulerian systems.** In the previous response, we discussed how MAVEN can be applied to physical systems involving thin-shell geometries and Euler formulations, and we provided a concrete example of how such an adaptation can be implemented.
> 5. **Detailed justification for translational and permutation invariance.**  Given a set of points \(x_1, x_2, \ldots, x_N\), computing their centroid ***c*** defines a **translation-equivariant operator**. Using ***c*** as the origin, we construct a local coordinate system for the points \(x_1, x_2, \ldots, x_N\), which is **translation-invariant by definition**. To ensure a unique and consistent ordering of the points under any translation, we sort the nodes using **translation-invariant features**. In our implementation, the feature used for sorting is **the distance from each point** to the centroid ***c***. With this procedure, both the node ordering and the node coordinates fed into the MLP remain uniquely determined and invariant under translation and permutation.
> 6. **Optimizable parameters. **In **Table 4 of Appendix D**, we provide the detailed parameter counts for all models. In practice, the only parameter-level difference between MAVEN and MGN lies in the **option for nodes of different types (vertex, facet, and cell) to use different feature dimensions**. We experimented with various combinations of feature dimensions while keeping the total parameter count fixed, and the results indicated that using a **shared feature dimension yields the best performance**. Therefore, we adopt a unified feature dimension for all node types in MAVEN.
>
> We thank the reviewer once again for the constructive feedback. If there are any further questions or points that would benefit from additional clarification, we would be happy to address them.
>
> [1]Sanchez, Jose, et al. "Robotic manipulation and sensing of deformable objects in domestic and industrial applications: a survey." *The International Journal of Robotics Research* 37.7 (2018): 688-716.
>
> [2]Pfaff, Tobias, et al. "Learning mesh-based simulation with graph networks." *International conference on learning representations*. 2020.
>
> [3]Yu, Youn-Yeol, et al. "Learning flexible body collision dynamics with hierarchical contact mesh transformer." *arXiv preprint arXiv:2312.12467* (2023).
>
> [4]Horie, Masanobu, and Naoto Mitsume. "Physics-embedded neural networks: Graph neural pde solvers with mixed boundary conditions." *Advances in Neural Information Processing Systems* 35 (2022): 23218-23229.
>
> [5]Bocheng Zeng, et al. PhyMPGN: Physics-encoded message passing graph network for spatiotemporal PDE systems. In *The Thirteenth International Conference on Learning Representations*, 2025.
>
> [6]Li, Zongyi, et al. "Fourier neural operator for parametric partial differential equations." *arXiv preprint arXiv:2010.08895* (2020).
>
> [7]Li, Zongyi, et al. "Fourier neural operator with learned deformations for pdes on general geometries." *Journal of Machine Learning Research* 24.388 (2023): 1-26.

---

### Official Review · Reviewer_2viX · 2025-10-30

**Soundness:** 3
**Presentation:** 3
**Contribution:** 2
**Rating:** 2
**Confidence:** 4

**Summary:**

This paper proposes a model that integrates volumetric geometric information into a GNN architecture to handle dynamic deformation simulation of meshes. Instead of using only vertices, as in conventional methods, the model integrates geometric features obtained from facets and cells and proposes a method for processing these elements. By incorporating this geometric information, the model demonstrated the advantage of achieving good performance in sparse meshes compared to other baselines.

**Strengths:**

I agree that the existing baselines failed to adequately capture the volumetric geometric information of the mesh. It is a strength that they proposed an architecture capable of containing all information that can be extracted from all possible constituent elements of the mesh: vertex, facet, and cell.

The architecture, designed to be applicable regardless of whether the mesh is tetrahedral or hexahedral, is a good foundation for generalization.

**Weaknesses:**

1. The biggest concern is that the computational cost for calculating the surface area, volume, and perimeter of cells and facet at every step seems to be vast for a fine mesh. Please provide the computational cost of the process of calculating geometric information based on the number of nodes and edges.

2. It was stated that facets within the search radius r are found using BVH when searching for contact surfaces. If this is the case, there is a need to specifically explain how the problem illustrated in Figure 1(d) is solved even in a sparse mesh. This seems to be the biggest differentiator from node-based methods, but I do not clearly understand how it is resolved.

3. If we assume that all facets in a sparse mesh within the radius r are in contact, a different problem from node-based methods might arise: since the model recognizes contact even when the facets overlap only slightly, wouldn't this lead to the issue of perceiving a wider contact area than the actual contact surface? A case study on how MAVEN resolves the issue shown in Figure 1(d) seems necessary.

4. The ablation study is conducted at too coarse a level, making it difficult to figure out what is effective for the model's performance. That is, it is difficult to clearly confirm whether the proposed model performs better because it actually incorporates more geometric information. Is the complex aggregator utilized superior to a simpler method of merely averaging and including the surrounding cell and facet information in the node features?

5. Long-range interaction is one of the important problems for GNN-based simulators. However, this methodology does not seem to include a discussion on that aspect.

**Questions:**

See Weaknesses.

---

> ### Author Response · Authors · 2025-11-23
> **Response 1**
>
> Thank you for your careful and constructive review. Your reviews have helped us strengthen both the clarity and technical quality of the manuscript. Below, we address each of the points you raised in detail.
>
> > computational cost for calculating the surface area, volume, and perimeter of cells and facet at every step seems to be vast for a fine mesh.
>
> The reviewer raised concerns about MAVEN’s computational efficiency. As reported in Appendix E, MAVEN runs about **23.89% slower** than the fastest baseline MGN[1], but achieves higher accuracy; compared with hierarchical models such as HCMT[2], it **reduces runtime by 43.80%** while maintaining better performance. MAVEN also requires only **3.30%** of the cost of traditional numerical solvers. These results show that MAVEN offers **acceptable computational efficiency**.
>
> Further, We analyzed runtime cost of MAVEN and across the three datasets, and the results are summarized in the table below.
>
> | Dataset         | Inference time | Cal. Feature time | Portion |
> | :-------------- | -------------- | ----------------- | ------- |
> | Deforming Plate | 48.59ms        | 7.87ms            | 16.20%  |
> | Cavity Grasping | 25.62ms        | 3.94ms            | 15.38%  |
> | Metal Bending   | 23.48ms        | 8.97ms            | 38.27%  |
>
> We observe that for the tetrahedral-mesh datasets (DP and CG), geometric feature computation accounts for roughly **16%** of the total runtime, increasing modestly as mesh density grows. Since *Deforming Plate* represents the dataset with fine mesh resolution in our experiments, these results indicate that the computational cost on tetrahedral meshes remains well controlled.
>
> For hexahedral meshes, computing exact cell volumes requires subdividing each hex into multiple tetrahedra, which introduces substantial overhead. **An alternative is to use Gauss quadrature[3] for hexahedral elements**, where the cell volume is approximated using the centroid. We follow classical FEM to employ an 8-point scheme to achieve second-order accuracy. This approximation trades a small amount of accuracy for a significant gain in inference speed.
>
> |           | Regular | Gauss quadrature |
> | :-------- | :------ | ---------------- |
> | Pos       | 810.42  | 869.38           |
> | Stress    | 4776.72 | 5032.67          |
> | Time cost | 23.48ms | 21.51ms          |

---

> ### Author Response · Authors · 2025-11-23
> **Response 2**
>
> >  As biggest differentiator from node-based methods, a case study on how MAVEN resolves the issue shown in Figure 1(d) seems necessary.
>
> We are pleased to further clarify the role of facets in accurately capturing contact behavior on sparse meshes. First, it is important to emphasize that MAVEN incorporates two key high-dimensional geometric structures: **cells** and **facets**. As described in **Lines 211–240 of the original manuscript**, MAVEN’s cell–facet framework integrates interaction signals on facets with volumetric geometric features in cells to achieve comprehensive and accurate information propagation in 3D continua. Cells and facets **play equally essential and complementary roles**, where facets capture critical contact and interaction information, while cells provide complete geometric and volumetric context.
>
> Next, we describe the differences between facet–facet contact and node–node contact from three perspectives: the **definition of contact**, the **variability of contact distances**, and the **validity of contact candidates**.
>
> 1. Mechanical contact is widely recognized as an interaction defined over a **two-dimensional surface region**, rather than at isolated points. [7] For this reason, **facet–facet contact formulations** provide a more physically meaningful and robust description of contact behavior. Even when two facets **overlap only slightly**, having access to the full geometric information of both surfaces allows us to **compute the potential contact area** and thus evaluate the correct contact response. In contrast, node-based methods cannot determine the continuous contact region around a single point and therefore cannot accurately estimate contact effects. For this reason, modeling contact at the facet level is a natural choice and is widely adopted in traditional numerical methods such as FEM.
>
> 2. In GNN-based methods, given a contact probing radius $R$, we aim to identify all regions whose distance is below $R$. However, under sparse mesh, the distance between two surface regions can differ significantly from the distance between their nodes. For example, consider two potentially contacting surfaces $[(0, 0, 0), (0, 1, 0), (1, 1, 0), (1, 0, 0)]$and $[(0.5, 0.5, 0.2), (0.5, 1.5, 0.2), (1.5, 1.5, 0.2), (1.5, 0.5, 0.2)]$. Although the minimum **surface-to-surface distance is 0.2**, the minimum distance between any pair of nodes is **no less than 0.73.** FIGNet[8] also highlights related scenarios in which **two cubes may collide via a corner hitting the other’s face, or an edge hitting the other’s edge**. Therefore, under the same contact detection radius, facet-based contact reliably identifies all valid contact pairs, whereas node-based contact fails to detect them correctly.
>
> 3. To ensure fairness in practice, we assign node-based methods **a larger probing radius** $R_W > R_F$, allowing them to detect a **comparable number of contact candidates** to the facet-based approach. However, increasing the radius also **brings in many spurious pairs that do not satisfy the actual contact conditions**, which can disrupt accurate contact prediction.  We define the set of valid contacting facet pairs as  $\mathcal{F}_C = \\{(F_1, F_2)\mid \text{dist}(F_1, F_2) \le R_F\\}$.
>
>    The corresponding valid node pairs are  $\mathcal{F}_N = \\{(u, v)\mid \exists\, F_1, F_2,\; u \in F_1,\; v \in F_2,\; (F_1, F_2) \in \mathcal{F}_C\\}$. We use the MBD dataset as a case study, where each frame contains on average **189.56** facet contact pairs and **720.11** node contact pairs. Our simplified BVH detection introduces some pairs with distance greater than $R_F$. Among them, the truly valid facet contacts average **150.79** pairs per frame, accounting for **79.55%** of all detected pairs. However, among all detected node pairs, only **117.23** belong to the **764.55** valid node-contact pairs, accounting for just **15.33%**, meaning most detected node pairs do not correspond to actual contact.
>
> Overall, these results demonstrate that facet-based contact provides a **physically meaningful, geometrically reliable, and highly efficient** way to identify true contact interactions on sparse meshes, whereas node-based contact suffers from severe detection inconsistencies and a high rate of spurious pairs. We hope this clarification fully addresses the reviewer’s concerns, and we would be happy to provide further details if needed.

---

> ### Author Response · Authors · 2025-11-23
> **Response 3**
>
> > The ablation study is conducted at too coarse a level.
>
> We appreciate your concern regarding the completeness of the ablation studies, as this has helped us further improve the thoroughness of our experimental evaluation.
>
> In our original ablation study, we included two variants: **Model A**, which replaces geometry-based aggregation coefficients with degree averaging to test the **role of geometric aggregation**, and **Model B**, which zero-pads geometric features of cells and facets to evaluate the **importance of explicit geometric inputs**.
>
> Following the reviewer’s suggestion, we added **Model C**, which removes explicit cell and facet nodes by averaging their precomputed geometric features onto neighboring vertex nodes and propagating them through standard message passing, to evaluate the role of explicitly representing higher-dimensional geometric elements. We also provided **ablation results on the Cavity Grasping dataset.** These results have also been incorporated into the revised version (Section 4.2).
>
> | Model   | Rollout  | CG Pos        | MBD Pos          | MBD Stress         |
> | ------- | -------- | ------------- | ---------------- | ------------------ |
> | Ours    | 50 / ALL | 5.21 / 15.41  | 276.73 / 810.42  | 4901.56 / 4776.72  |
> | Model A | 50 / ALL | 6.26 / 17.45  | 402.79 / 926.71  | 5302.67 / 6683.94  |
> | Model B | 50 / ALL | 5.41 / 15.933 | 391.40 / 1652.31 | 6839.26 / 6680.39  |
> | Model C | 50 / ALL | 6.05 / 17.08  | 632.71 / 1680.20 | 9802.55 / 10375.86 |
>
> The table above presents the results of our ablation study. We observe that replacing geometry-aware aggregation with simple averaging (Model A) performs significantly worse on the generally sparse CG dataset. This indicates that **such sparsity requires capturing detailed intra-element geometry** to surpass standard node-based methods. Similarly, omitting explicit geometric features (Model B) leads to severe degradation on the highly sparse MBD dataset. This suggests that in extremely sparse settings, traditional GNNs **struggle to infer local geometric structure implicitly** and therefore fail to accurately capture the underlying geometric topology. For Model C, which does not explicitly model high-dimensional geometric elements, its performance on both datasets is close to traditional node-based methods. This suggests that adding geometric features alone, without modeling higher-dimensional topology, is **insufficient for nodes to fully capture surrounding geometric structure.**
>
> These ablation results collectively demonstrate the effectiveness and necessity of MAVEN’s design choices, including the **explicit modeling of higher-dimensional geometric elements**, the **explicit computation of geometric features**, and the use of **geometry-aware aggregation** for message updates.

---

> ### Author Response · Authors · 2025-11-23
> **Response 4**
>
> > This methodology does not seem to include a discussion on long-range interaction.
>
> As discussed in **Section 3.6 of the original manuscript**, MAVEN is primarily designed as a geometry-based local operator approximator, similar in spirit to MGN. Below, we provide a more detailed explanation of the design principles underlying MAVEN.
>
> - **MAVEN is fundamentally centered on modeling and exploiting high-dimensional geometric structures within the mesh.** It is important to emphasize that the core contribution of MAVEN lies in its **explicit modeling of high-dimensional geometric structures in the mesh**, namely facets and cells. By incorporating precisely computed geometric features and introducing a facet–cell graph structure to better capture geometric and topological relationships, MAVEN leverages information that is **typically overlooked by existing local GNN methods** such as MGN and by **hierarchical methods** such as HCMT that focus on long-range interactions. Both the quantitative results and visualizations demonstrate that MAVEN achieves lower error and produces more stable geometric results.
> - **MAVEN can be easily extended to capture long-range interaction.** Existing GNN methods for capturing long-range information typically expand each node’s receptive field through pooling strategies such as kNN-based clustering[4], BFS grouping[5], or global slicing[6]. Since these techniques are **generally applicable to arbitrary graphs**, as noted in **Lines 376–377 of the original manuscript**, they can be **directly applied to the cell–facet graph** in MAVEN to support long-range information propagation. However, such pooling operations often introduce substantial computational overhead (as observed in HCMT), and they offer **limited benefits for capturing global high-dimensional geometric structure**. Therefore, they are not the focus of our work, which centers on the explicit modeling of mesh-based geometric information.
>
> For these reasons, we design MAVEN as a local operator that focuses on capturing and leveraging high-dimensional geometric information within the mesh.  For long-range interactions, we also plan to develop a geometry-aware hierarchical extension as an important direction for future work, and this has been added to the revised version of the manuscript(Appendix G).
>
> We sincerely appreciate the reviewer’s careful assessment and valuable input. Your suggestions have been highly helpful in refining the paper. If there are any remaining questions or points that would benefit from further discussion, we would be happy to address them.
>
> [1]Pfaff, Tobias, et al. "Learning mesh-based simulation with graph networks." *International conference on learning representations*. 2020.
>
> [2]Yu, Youn-Yeol, et al. "Learning flexible body collision dynamics with hierarchical contact mesh transformer." *arXiv preprint arXiv:2312.12467* (2023).
>
> [3]Hughes, Thomas JR. *The finite element method: linear static and dynamic finite element analysis*. Courier Corporation, 2012.
>
> [4]Janny, Steeven, et al. "Eagle: Large-scale learning of turbulent fluid dynamics with mesh transformers." *arXiv preprint arXiv:2302.10803* (2023).
>
> [5]Cao, Yadi, et al. "Efficient learning of mesh-based physical simulation with bi-stride multi-scale graph neural network." *International conference on machine learning*. PMLR, 2023.
>
> [6]Lei, Bo, Victor M. Castillo, and Yeping Hu. "Physically Aligned Hierarchical Mesh-based Network for Dynamic System Simulation." Openreview, 2025.
>
> [7]Lu, Jia. "Isogeometric contact analysis: Geometric basis and formulation for frictionless contact." *Computer Methods in Applied Mechanics and Engineering* 200.5-8 (2011): 726-741.
>
> [8]Kelsey R. Allen, et al. Learning rigid dynamics with face interaction graph networks. *arXiv preprint* arXiv:2212.03574, 2022.

---

> > ### Comment · Reviewer_2viX · 2025-11-27
> >
> > Thank you for your rebuttal.
> >
> > I think my major concerns are well addressed, and Model 4 in the ablation study shows the superiority of the MAVEN architecture design.
> >
> > Regarding the computational cost, since the authors explain it using experimental times rather than Big O notation, it is not theoretically clear, but I think this is an acceptable alternative for resolving my concern.
> >
> > I will raise my score to 6. Good luck.

---

> > > ### Author Response · Authors · 2025-11-27
> > >
> > > Thank you for your recognition of our work, and for the thoughtful and constructive feedback you provided. Your insights were very helpful in strengthening the paper. We truly appreciate the time and care you devoted to the review.

---

### Official Review · Reviewer_31zk · 2025-11-05

**Soundness:** 3
**Presentation:** 3
**Contribution:** 2
**Rating:** 6
**Confidence:** 4

**Summary:**

This paper proposes a mesh-aware volumetric encoding network to predict physically meaningful 3D deformation. The paper shows better performance compared with alternative baseline models.

**Strengths:**

The overall idea of encoding meshes at different levels is plausible.

The performance improvements compared with existing methods.

**Weaknesses:**

The deformation evaluated is somewhat limited. For more diverse materials and object shapes, it would be useful to show the generalizability.

The proposed method essentially relies on volumetric information, but it is unclear whether the compared methods are surface or volume based. This leads to questions regarding whether the evaluation is fair.

**Questions:**

As all the examples shown have limited variations for the geometry, does the method generalize to more flexible shapes?

---

> ### Author Response · Authors · 2025-11-23
> **Response 1**
>
> We sincerely appreciate the reviewer’s thorough assessment and constructive remarks. Your feedback has contributed meaningfully to improving the clarity and rigor of the manuscript. In the following, we provide detailed responses to each of the points you raised.
>
> > Evaluation on more diverse materials and object shapes
>
> Thank you for the insightful comment. We fully agree that evaluating on more datasets is generally desirable. However, we would like to clarify that the limited number of datasets is due to the **scarcity of publicly available benchmarks** that contain **well-shaped high-dimensional geometric topologies** such as cells and facets. This information is a core requirement of MAVEN rather than an arbitrary choice made in our experimental design.
>
> The key idea behind MAVEN is to **leverage intrinsic mesh-based geometry**, including cell and facet level structures. These higher-order geometric components are essential for capturing spatial relationships that graph-only representations cannot reliably encode. As a result, meaningful evaluation of MAVEN requires datasets with consistent and high-quality geometric structures.
>
> In practice, most existing graph or physics-simulation datasets only **provide node-edge connectivity and do not include volumetric meshes and cell decompositions**. Applying MAVEN to such datasets would require reconstructing volumetric mesh structures from incomplete geometric information. This reconstruction is technically very challenging and often infeasible, as **high-quality volumetric mesh generation from point cloud is itself a difficult and largely unsolved problem**.
>
> For these reasons, we focus on datasets that either provide high-quality meshes or allow reasonably accurate geometric recovery. Specifically, we use Deforming Plate, which provides native meshes; Cavity Grasping, for which we carefully reconstructed a faithful volumetric mesh based on the available geometric signals; and a hexahedral Metal Bending dataset that we generated ourselves to address the lack of publicly available datasets featuring **elastoplastic behavior and structured hexahedral meshes**. These datasets collectively span diverse geometric configurations, including tetrahedral and hexahedral meshes, and cover a range of physical tasks and deformation behaviors. Under the requirement for accurate and **high-quality** geometric information, they represent a comprehensive and practically usable set of benchmarks currently available.
>
> We hope that these explanations satisfactorily address the reviewer’s concern. In future work, we intend to broaden and publicly release our datasets in order to facilitate progress in this direction(added to Appendix G in revision version). We would also appreciate any recommendations the reviewer may have regarding high-quality datasets involving elastoplastic deformation or hexahedral meshes.
>
> > It is unclear whether the compared methods use surface-based or volume-based representations, raising questions about the fairness of the evaluation.
>
> Regarding the reviewer’s concern about the fairness of comparing MAVEN with baselines that differ in their use of facet and cell information, we would like to clarify that **Sections 2 and 3.6** in the original manuscript provide an overview of how the compared methods are categorized and how they relate to our approach. Here, we provide a brief summary of how each method mentioned in the paper **utilizes geometric elements such as cells and facets**.
>
> - For the **basic GNN methods** (MGN [1] and Graph Transformer [2]) and the **hierarchical variants** (HOOD [3] and HCMT [4]), these approaches rely solely on vertex-based representations. As noted in Lines 153–155 of the original manuscript, they only model mesh vertices and therefore **do not make use of the higher-dimensional geometric elements** present in the mesh.
> - **FIGNet**[5] focuses on rigid-body contact and uses **only facet-level information**, without modeling volumetric propagation inside objects. In contrast, MAVEN incorporates both facets and internal cells, enabling accurate representation of intra-object dynamics, which leads to significantly improved performance in our experiments.
>
> To ensure a fair comparison across all methods, we **matched the parameter budgets** for each model and trained them using exactly **the same hyperparameter settings** (as reported in **Appendix D** of the original manuscript). For contact modeling, we adjusted the probing radius so that facet-based and point-based methods **detect a comparable number of contact candidates**. In addition, all models were evaluated under **similar inference-time budgets** (as shown in **Appendix E **of the original manuscript). With these controls in place, we believe that the comparison is conducted under fair and consistent conditions, and the reported results are therefore reliable and meaningful.

---

> ### Author Response · Authors · 2025-11-23
> **Response 2**
>
> > Does the method generalize to more flexible shapes?
>
> In response to your concerns regarding the geometric settings to which MAVEN can be applied, we are pleased to clarify and restate the scope of geometries that MAVEN is designed to support below.
>
> - **MAVEN is compatible with meshes composed of arbitrary element types.** In our experiments, we demonstrate that MAVEN performs well on tetrahedral and triangular-surface meshes (Deforming Plate[1] and Cavity Grasping[6]) as well as hexahedral and quadrilateral-surface meshes (Metal Bending). These element types constitute the **majority of geometric primitives used in practical engineering applications**. Furthermore, MAVEN can be applied to any mesh elements for which surface area and volume can be computed, including hybrid meshes composed of mixed element types.
> - **MAVEN is applicable to arbitrary geometries that can be discretized into meshes.** In our experiments, we show that MAVEN performs well on **slender structures** (Metal Bending), **hollow shell-like geometries** (Cavity Grasping), and **plate-like shapes** (Deforming Plate), all of which are common in engineering applications. More broadly, we expect MAVEN to support most geometric shapes for which high-quality mesh discretization can be obtained.
>
> Overall, MAVEN demonstrates strong performance across commonly used mesh element types and geometric shapes. We believe that MAVEN can be further extended to a wider range of flexible shapes, provided that high-quality mesh discretizations can be obtained for those geometries.
>
> We thank the reviewer once again for the insightful feedback and constructive suggestions. Your comments have helped us substantially improve the clarity and rigor of the manuscript. We would be glad to provide any further clarification or discuss additional questions if needed.
>
> [1]Pfaff, Tobias, et al. "Learning mesh-based simulation with graph networks." *International conference on learning representations*. 2020.
>
> [2]Yun, Seongjun, et al. "Graph transformer networks." *Advances in neural information processing systems* 32 (2019).
>
> [3]Yu, Youn-Yeol, et al. "Learning flexible body collision dynamics with hierarchical contact mesh transformer." *arXiv preprint arXiv:2312.12467* (2023).
>
> [4]Grigorev, Artur, Michael J. Black, and Otmar Hilliges. "Hood: Hierarchical graphs for generalized modelling of clothing dynamics." *Proceedings of the IEEE/CVF conference on computer vision and pattern recognition*. 2023.
>
> [5]Kelsey R. Allen, et al. Learning rigid dynamics with face interaction graph networks. *arXiv preprint* arXiv:2212.03574, 2022.
>
> [6]Linkerhägner, Jonas, et al. "Grounding graph network simulators using physical sensor observations." *arXiv preprint arXiv:2302.11864* (2023).

---

### Author Response · Authors · 2025-11-23
**Revision**

We would like to express our sincere appreciation to the reviewers, as well as the ACs, SACs, and PCs, for their time, effort, and insightful feedback. In response to these comments, we have refined the manuscript to enhance its clarity and overall quality. The main revisions are summarized as follows:

1. Advised by reviewer ZTpm, we have added a description to the caption of the main flowchart(Figure 2) and corrected several issues related to improper wording (Section 3.3).
2. Advised by reviewer 3Wkt, we have added a comparison with existing methods that leverage high-dimensional geometry in the discussion subsection (Section 3.6).
3. Advised by reviewer 2viX, we have added additional ablation study results (Section 4.2), providing further insights into the design of the model modules.
4. Advised by reviewer 31zk, 2viX and ZTpm, we have added a detailed discussion (Section G) addressing the suggestions on extending applicability and clarifying handling of long-range interactions.

We use blue text to highlight the modification. We hope these revisions address the reviewers’ concerns. We also look forward to further in-depth and detailed discussions, which will undoubtedly help us continue improving the quality of our work.

---

### Author Response · Authors · 2025-11-27
**Regarding the Ongoing Discussion Phase**

Dear Reviewers,

We hope this message finds you well. As the discussion phase is approaching its conclusion in **less than a week,** we would like to kindly check whether there are any further comments or clarifications you would like us to address. Your feedback is greatly appreciated, and we are committed to resolving any remaining questions to further strengthen our manuscript.

 Thank you very much for your time and effort in reviewing our paper.

---

### Meta-Review · Area_Chair_MzLk · 2025-12-29

**Summary:**

This paper introduces MAVEN, a mesh-aware volumetric encoding network for simulating 3D flexible deformation that explicitly models higher-dimensional geometric elements (3D cells and 2D facets) beyond the standard vertex-edge representations used in existing GNN-based simulators. The approach incorporates geometric features such as volume, surface area, and perimeter into the model through position-aware aggregators and propagates information via a cell-facet graph structure using two-stage message passing.

The paper received scores of 6, 6, 6, and 2 (initially), with Reviewer `2viX` raising their score from 2 to 6 after rebuttal. The revised median score is 6 (marginally above acceptance threshold). All reviewers acknowledged the core contribution of incorporating higher-dimensional geometric elements, though concerns were raised about computational costs, ablation completeness, generalizability, and applicability beyond volumetric solids. The authors provided comprehensive rebuttals addressing computational efficiency, additional ablations, comparison fairness, and extensions to other geometric regimes.

**Reviewer Concerns:**

### Addressed Concerns:
1. **Computational Cost (`2viX`)**: Authors demonstrated that MAVEN runs only 23.89% slower than MGN but achieves higher accuracy, and is 43.80% faster than HCMT. Geometric feature computation accounts for 16-38% of total runtime depending on mesh type. An approximate Gauss quadrature scheme was provided for hexahedral meshes to further reduce costs.

2. **Ablation Studies (`2viX`)**: Authors added Model C to the ablation study, which removes explicit cell and facet nodes by averaging geometric features onto vertices. Results showed Model C performs similarly to node-based methods, confirming that explicit modeling of higher-dimensional elements is essential. Additional experiments on the Cavity Grasping dataset further validated the design choices.

3. **Facet Representation Necessity (`2viX`)**: Authors clarified that facets provide physically meaningful contact representation (surface-to-surface vs point-to-point), reduce distance variability under sparse meshes, and yield valid contact pairs (79.55% vs 15.33% for node-based detection in the MBD dataset).

4. **Long-Range Interactions (`2viX`)**: Authors explained MAVEN focuses on local geometric modeling and can be extended with standard pooling strategies (kNN clustering, BFS grouping) applied to the cell-facet graph, though this is left for future work.

5. **Comparison Fairness (`31zk`)**: Authors confirmed all models were evaluated under matched parameter budgets, identical hyperparameters, similar inference-time budgets, and comparable numbers of detected contact candidates.

6. **Generalizability (`31zk`)**: Authors clarified MAVEN applies to arbitrary mesh element types (tetrahedral, hexahedral, mixed) and geometries (slender, hollow, plate-like) provided high-quality discretization is available. The scarcity of public datasets with high-quality cell/facet geometry limits broader evaluation.

### Outstanding Concerns:
1. **Applicability Beyond Volumetric Solids (`ZTpm`)**: While authors explained MAVEN can conceptually adapt to thin-shell geometries and Eulerian formulations with minor modifications, the method primarily targets 3D volumetric solids. The need for geometry-aware adjustments reflects MAVEN's higher geometric fidelity but limits immediate applicability to other regimes.

2. **Dataset Diversity (`31zk`)**: Evaluation is limited to three datasets due to scarcity of publicly available benchmarks with high-quality cell/facet geometry. This constrains assessment of generalizability across diverse materials and shapes.

3. **Comparison with Geometry-Based Methods (`3Wkt`)**: While authors added discussion comparing MAVEN with PhyMPGN and FIGNet in the revision, reviewer `3Wkt` expressed limited domain expertise and deferred to other reviewers.

4. **PINNs Relationship (`ZTpm`)**: Authors explained PINNs are unsuitable baselines due to requirements for explicit PDE formulations, instance-specific training, and regular-domain assumptions that differ fundamentally from MAVEN's data-driven, generalizable learning setting.

**Reviewer Scores:**

**Initial Scores:**
- **Reviewer `31zk`**: 6 (marginally above acceptance threshold)
- **Reviewer `2viX`**: 2 (reject) → 6 (marginally above acceptance threshold, raised after rebuttal on 26 Nov 2025)
- **Reviewer `ZTpm`**: 6 (marginally above acceptance threshold)
- **Reviewer `3Wkt`**: 6 (marginally above acceptance threshold, limited domain expertise)

**Expected Post-Discussion Scores:**
- **Reviewer `31zk`**: 6 (likely to remain)
- **Reviewer `2viX`**: 6 (confirmed after rebuttal)
- **Reviewer `ZTpm`**: 6 (likely to remain)
- **Reviewer `3Wkt`**: 6 (likely to remain, defers to others)

**Median Score:** 6 (marginally above acceptance threshold)

---

### Decision · Program_Chairs · 2026-01-26

Accept (Poster)